

# Calibrating a process-based forest model with a rich observational dataset at 22 European forest sites.

David Cameron[1], Christophe Flechard[2], and Marcel Van Oijen[1]

[1]Centre for Ecology and Hydrology, Bush Estate, Penicuik, Midlothian, EH26 0QB, United Kingdom
[2]INRA, Equipe Agrohydrologie, UMR INRA/Agrocampus 1069 SAS, 65, rue de St-Brieuc, CS 84215, 35042 Rennes Cedex, France

**Correspondence:** David Cameron (dcam@ceh.ac.uk)

**Abstract.** In recent years model-data interaction has improved through use of probabilistic techniques to inform and reduce the uncertainty of model parameters, while also taking into account observational uncertainty. This study builds on previous work, through access to a richer representation of the plant-soil ecosystem at multiple European forest sites, than was previously available. Given this rich dataset, we asked which observational datasets were most effective in reducing uncertainty in model

predictions and model-data differences. Also, since there is a lack of consensus about whether it is more beneficial to calibrate forest sites separately or together we revisited this question with a particular emphasis on which is most effective in reducing model-data differences and uncertainty. We performed single dataset Bayesian calibrations (BC) and compared the results with a calibration with all the observations included. We also compared calibrations where each pine forest site was calibrated separately with a calibration where all the pine sites were calibrated together. While measurements of plant and soil carbon

stocks were more sparse, their inclusion in the BC were more important for reducing model-data differences and uncertainty in the above and belowground carbon pools than the greater numbers of carbon and water flux data. Our results suggest that use of calibration data representing just a few aspects of the ecosystem could be problematic, since improved model-data fits for the parts of the system represented by the data could be at the expense of other part of the system, where the model-data fit worsened. The single dataset calibrations helped to diagnose where there may be inconsistencies between different datasets or

between the model and data or both. These inconsistencies hampered the reduction in model-data differences in the calibration with all the observations present. As expected, we found a strong relationship between the quantity of data included in the calibration and the uncertainty reduction after BC, finding the largest reduction in uncertainty when all the observations were included. For some ecosystem variables uncertainty reduced after calibration but model-data differences increased. This would suggest that there were deficiencies in the model or systematic errors in the data or both. These results advocate the use of

calibration datasets which represent the rich diversity of the ecosystem under investigation but where model discrepancies and data systematic errors are explicitly represented in the BC. While separate calibrations at each forest site generally reduced model-data differences more than calibrating at all the sites together, parts of the ecosystem that were sparsely observed benefited more from the multi-site calibration. Multi-site calibration led to larger and more consistent reductions in uncertainty than separate calibrations at each site, especially for ecosystem variables with fewer observations. These results support the




use of Bayesian hierarchical calibration which allows variation in model parameters between different sites while allowing information to be shared across sites for sparsely observed ecosystem variables.

# 1 Introduction

Forests have a considerable influence on the global carbon, nitrogen and water cycles and northern hemisphere forests are
thought to have a dominant role in global terrestrial carbon sequestration. (Pan et al., 2011; Ciais et al., 2010)

Predicting the future of European forests is crucial both to quantify their continuing role in environmental change (Whitehead, 2011) and also to assess the impact of future environmental change on European forests and whether these changes are likely to pose risks to the future health of forests in Europe (Reichstein et al., 2013; Van Oijen et al., 2014).

While process-based forest models (PBMs) are an essential tool for quantifying the future of European forests significant
uncertainties remain in the predictions from these models. Many different modelling approaches and parametrisations exist with no one model having widespread applicability across Europe (Mäkelä et al., 2012). Uncertainties are present in the imperfect process representation and structure of the PBMs, initial states and model parameters as well as in driving meteorological input data (Reyer et al., 2016). Without quantifying these uncertainties it is hard to assess the value of PBMs for predicting the future of forests.

In recent years there has been increasing use of mathematical probability theory in quantifying and reducing parameter uncertainties in process-based ecological models (Van Oijen, 2017). The method, known as Bayesian Calibration (BC), also takes into account uncertainties in observational data which are likewise imperfect.

While initial Bayesian calibrations of ecosystem PBMs were made at single locations (Van Oijen et al., 2005, 2011) more recent studies have made calibrations at multiple ecological sites (Reinds et al., 2008; Lehuger et al., 2009; Van Oijen et al.,
2013; Van Oijen and Höglind, 2015; Molina-Herrera et al., 2015). These multi-site calibration studies recognise the danger that any individual observational site may be atypical and hence a calibration at a single site may not be useful for predictions at other sites or wider regions.

One question which a number of studies have posed is whether model performance is improved by calibrating at each observational site separately (unpooled) or all sites together (pooled). The implicit assumption in a pooled calibration is that model
parameters are generic and hence will have the same value at all locations whereas unpooled calibrations allows parameters to have different values spatially. The greater model flexibility accepted in unpooled calibrations is less useful than pooled calibrations for making predictions at unobserved sites or for the whole region.

Reinds et al. (2008) calibrated a soil ionic concentration model at 122 forest sites in Europe and found that unpooled calibrations performed better than the pooled calibration. Lehuger et al. (2009) calibrated a model of $N_2O$-emissions in crops
at 11 sites in France and also found that the unpooled calibrations performed best. They suggested that a compromise combining the strengths of pooled and unpooled calibrations might be to perform a hierarchical Bayesian calibration where parameters were allowed to vary spatially but are constrained by a probability distribution.





Studies which calibrated process-based forest models at multiple forest sites include Van Oijen et al. (2013). They calibrated six forest models at 12 sites in four countries with just height and diameter at breast height (DBH) data. The parameters were calibrated either for all sites together (pooled) or on a country by country basis (partially unpooled). They found that calibrating separately for each country did not significantly improve the with-in country predictions of the model versus the

pooled calibration. Molina-Herrera et al. (2015) calibrated the model LandscapeDNDC against NEE, GPP and respiration flux data at 10 forest sites. They found that the unpooled calibrations performed better than pooled and suggested that this was because parameters in the unpooled calibration were able to reflect the adaptation of the forest system to local environmental conditions. There was however, an issue of double-counting of observations in this study since any two of NEE, GPP and respiration equals the other. Further, GPP is a model derived quantity and so does not provide observations independent of

NEE and hence its inclusion is more akin to a comparison of LandscapeDNDC against the model used to partition GPP from NEE fluxes. This study also only included a subset of the model parameters from the plant growth submodel for calibration. This is potentially dangerous without first assessing which parameters are important in the BC since it can cause compensating errors in the posterior values of the calibrated parameters and may have disadvantaged the pooled calibration. Molina-Herrera et al. (2015) also did not propagate model parameter uncertainties to model predictions.

This study will extend previous multi-site forest model calibrations through inclusion of data on many more aspects of the forest ecosystem than in previous studies. The calibration includes observations of carbon stocks in the plant and soil, height, DBH and LAI as well as carbon fluxes from eddy covariance measurements and NO, $N_2O$ emissions and soil respiration from soil chambers. In addition, the calibration incorporates soil water content data and evapotranspiration flux data to help constrain the water cycle in the model. Including data that encapsulate a larger proportion of the forest ecosystem should constrain a

wider range of processes included in forest PBMs. Further, the number of forest sites represented in this study (22) is almost double that of previous PBM forest model calibrations.

Since this study includes such a relatively extensive and rich observational dataset (i.e. multiple ecosystem variables measured over time at many locations), a particular focus of this work is on the influence of the calibration on model predictions and in particular, which observational datasets were most effective in reducing uncertainty in model predictions and data-model

differences. Also given the lack of agreement in previous forest studies about whether pooled or unpooled calibration is more beneficial for reducing model-data differences we revisit this question for pine forests and also consider which calibration is most effective in reducing uncertainty in model predictions. Since model parameterisations are specific to models we will present results showing how model output uncertainty and model-data differences changed as a result of calibration rather than how the model parameter uncertainty changed as a result of the BC.




## 2 Methods and materials

### 2.1 Observational data

#### 2.1.1 Forest sites

The 22 forest sites that were chosen for the calibration of PBM model BASFOR were all relatively healthy, mature, mono-
cultural forest stands, with either broadleaf deciduous or evergreen needle trees. The sites are listed in Table 1. Table 2 lists
literature references that were used for the forest sites. A map showing the forest site locations can be found in the supplemen-
tary material.

#### 2.1.2 Calibration data

Soil water content (SWC) was measured hourly and continuously using Time Domain Reflectrometry (TDR) probes at typically
three depths (5, 20 and 50cm). Data was available for most sites. The data was averaged to daily values to match the timestep
of the model.

Atmospheric/surface carbon and water flux data were taken from the CarboEurope Integrated Project (CEIP) database, the
European Fluxes Database Cluster (http://gaia.agraria.unitus.it/home) or the GHG Europe portal (http://www.europe-fluxdata.
eu/ghg-europe).

Soil respiration, $NO_2$ and NO were measured in static chambers at the forest sites. Given the difficulty in collecting manual
soil measurements that closely match what is calculated in the model, we opted to create an annual integrated metric for these
soil flux data.

Data from CEIP and other project (e.g. FLUXNET) databases as well as publications (Table 2) were used for canopy height
(H), leaf area index (LAI), diameter at breast height (DBH), soil organic matter (CSOM), roots (CR), stems, branches, leaves
(CLBS) and litter (CLITT). Additional data for above and belowground carbon stocks were provided by the global database
assembled by Luyssaert et al. (2007).

#### 2.1.3 Soil depth and soil water retention data used to set prior ranges.

Measurements of wilting point, field capacity and rooting depth were available at a number of sites from the CEIP database
and also peer-review publications (Table 2) and are listed in Table 4 below. Saturation measurements were inferred from
the largest soil water content measurement recorded at each site. Where measurements or publications were not available or
could not be found, wilting point and field capacity were estimated using tabulated values from the German soil texture clas-
sification Ad-Hoc-AG Boden (2004), by linking soil texture, from the CEIP and other project (e.g. FLUXNET) databases,
or peer-reviewed publication, to characteristic values of soil water potential and volumetric soil water content by means of
pedo-transfer functions (van Genuchten, 1980) https://www.bgr.bund.de/DE/Themen/Boden/Netzwerke/Adhocag/Downloads/
Ergaenzungsregel_1_18.pdf?__blob=publicationFile&v=2. Where information on rooting depth from measurements or publi-
cations were not available a rooting depth of 1m was assigned.





## 2.2 Brief description of the model BASFOR

The BASic FORest simulator, BASFOR, (https://github.com/MarcelVanOijen/BASFOR) Van Oijen et al. (2005) is a deterministic forest model. The model simulates carbon and nitrogen cycling in trees, soil organic matter and litter. BASFOR is built from well known process representations. Light absorption is calculated by Beer's law. GPP is calculated as light absorption
times a light-use efficiency (LUE). NPP is calculated as a fixed ratio of GPP. LUE is temperature-, $CO_2$ and soil water content-dependent and may be reduced if insufficient nitrogen is taken up by the plants. Potential nitrogen uptake scales with root system surface area. Actual nitrogen uptake is the minimum of demand, determined by tissue N-concentration, and potential uptake. Allocation of assimilates follows allometric rules, but water stress may limit leaf area index (LAI). Turnover of tree and soil components proceeds at temperature-dependent relative rates. BASFOR has a daily timestep. The model structure was
described by Van Oijen et al. (2005). Papers describing more recent model developments are Van Oijen and Thomson (2010) and Van Oijen et al. (2011).

## 2.3 Model driving/input data

### 2.3.1 Weather data

BASFOR requires the following weather input variables: Mean daily temperature, precipitation, radiation, vapour pressure
and 2m wind speed. These weather data were obtained from in-situ measurements above each forest site. Since weather data were not available back to the year of planting, the data were replicated backwards in time based on the available time series (typically 5 to 20 years).

### 2.3.2 Management

Present day stand densities were available for all sites from the CEIP and other project databases such as FLUXNET and
complemented with data from publications (Table 2). However, a reasonably complete time-series of planting densities and stand thinning events from planting were available for only a few sites. Where planting density was unavailable, we assumed a default initial value of 4500 trees ha$^{-1}$. For sites with missing thinning histories, we created plausible thinning histories using stand density and thinning data where available and filling in the gaps by loosely following the principle in Cameron et al. (2013) that after an initial thinning of 40% at 20 years there would be decadal thinning of 20%. A figure showing the stand
history reconstructions used is given in the supplementary material.

### 2.3.3 Atmospheric nitrogen deposition

Historical nitrogen deposition timeseries inputs to BASFOR were created by assuming that sites followed the same relative time course from planting until 2005-2010 following Van Oijen et al. (2008) with a scaling for each site using present-day estimates from Flechard et al. (2011) and summarised below. N deposition was assumed to rise from a base value below 0.5
g N m$^2$ yr$^{-1}$ at the turn of the 20th century, increasing sharply after World War II, peaking in the early 1980s and decreasing




by approximately one third until 2005-2010. A graph showing the historic N deposition time-series used for each site can be found in the supplementary materials.

A detailed description of how nitrogen deposition time-series were estimated for each site after 2005-2010 can be found in Flechard (in prep.) with a brief summary presented here. The dry deposition fraction of the total N deposition was estimated from concentrations using an ensemble average of four different deposition models (Flechard et al., 2011). Wet deposition data came from three sources. Firstly a chemical analysis of precipitation at each of the sites. Secondly through spatial interpolation (kriging) of data from the wet deposition monitoring network in Europe to the sites. Finally wet deposition was calculated by the 50x50 km EMEP model. The three data sources were combined by taking the arithmetic mean. Wet and dry deposition was then added together to give total N deposition values.

### 2.3.4 Soil nitrogen at planting

Total soil nitrogen values at planting were not estimated in the calibration and were taken from Table (1) p70. of Van Oijen et al. (2008).

### 2.3.5 CO$_2$ time-series

The historical time-series of increasing CO$_2$ was the same as that used in Van Oijen et al. (2014) and is based on data from ice-core records and NOAA atmospheric observations for the years 1901–2010.

### 2.4 Bayesian calibration methodology

In Bayesian model parameter calibration our aim is to infer the joint probability distribution for model parameters values given the observational data. This is written as $P(\theta|D)$, where $\theta$ is the parameters and D is data and can be calculated using the probability calculus as

$$P(\theta|D) \propto P(D|\theta)P(\theta) \qquad (1)$$

Where $P(\theta|D)$, $P(D|\theta)$ and $P(\theta)$ are known as the posterior, likelihood and prior respectively.

### 2.4.1 Prior distribution and methodology

The prior probability distribution represents the state of our knowledge about the correct parameter values prior to the comparison against the observational data. We chose the Beta distribution for the prior probability distribution, which is bounded and can be non-symmetric.

Given our choice of the Beta distribution we needed to specify a mode, maximum and minimum for each of the parameters that we calibrated. Since we did not possess reliable prior knowledge about correlations between parameters, we did not specify any covariance structure in our prior. Correlations between parameters were quantified in the joint posterior distribution (not shown).





With a few exceptions most of the parameters of BASFOR are not known very precisely, therefore, we chose to calibrate nearly all of the parameters in BASFOR. Following this approach avoids arbitrarily setting the value of any parameter and underestimating uncertainty.

While it was likely that not all the parameters would be informed by the data, without using a screening procedure beforehand we did not know which parameters these would be. Since BASFOR is relatively inexpensive to run there was no need to reduce the number of parameters included in the calibration.

Since we were calibrating the parameters for multiple forest sites we needed to decide whether the parameters should be considered to be generic across all sites or whether parameters should have site-specific values. In the development of BASFOR we chose a modelling methodology and structure of the model such that parameters should have general applicability to all European forests of the same type. We therefore chose to keep all parameters as generic within the categories pine, spruce and deciduous forests. The exceptions were parameters which defined the water retention curve and rooting depth and also initial soil and litter carbon values which were considered to be specific to each site. This was done by assigning a separate parameter for each site, for these site-specific parameters, which added to the number of parameters in the calibration.

A separate calibration was made for each of pine, spruce and deciduous forests.

### 2.4.2 Prior parameter minimum, mode and maximum

The prior values for the generic parameters are listed in the supplementary material. The same prior was used for pine and spruce. Initial soil litter and organic matter carbon had the same prior within each category of pine, spruce and deciduous forests.

Where possible, parameter priors were estimated from measurements at sites or from literature. In many cases this was not possible as either data were not available or the model parameter had a different role in the model than that measured. In these cases a wide prior was used. Values for these wide priors were guided by previous studies with BASFOR (Van Oijen et al., 2005; Van Oijen and Thomson, 2010; Van Oijen et al., 2011).

### 2.4.3 Soil depth and water retention parameters

Prior ranges for soil water retention and soil depth parameters were chosen on a site by site basis using site observations as described in section 2.1.3. These data were used to inform the modes of the soil water retention and rooting depth parameters at each site. Prior uncertainty was set at 20% of the mode value for measured quantities and 40% for estimated quantities. The soil water saturation parameter was assigned a prior uncertainty of 30% of the mode value. Tables showing the prior ranges for each site are given in the supplementary material.

### 2.4.4 Likelihood function

The likelihood quantifies the chances of obtaining the observational data from the model at a given parametrisation. The uncertainty about random data error has often been represented by independent Gaussian distributions for the observations. As





discussed in Van Oijen et al. (2011) this can overestimate the information content of the observations leading to an underesti-
mate in uncertainty. To help alleviate this issue we used the heavy-tailed distribution of Sivia (Sivia and Skilling, 2006).

$$P(D|\theta) = \prod_{i=1}^{n} \frac{1}{\sigma_i \sqrt{2\pi}} \frac{1 - exp(-R_i^2/2)}{R_i^2} \qquad (2)$$

where n is the number of data points, $\sigma_i$ is a measure of the uncertainty about random error of the ith data point, and $R_i$ is
the difference between model output and ith data point, divided by $\sigma_i$.

While BASFOR is a daily time-step model it is unlikely to give accurate predictions at this temporal resolution. We therefore
aggregated all the daily data (carbon and water fluxes and soil water content) to a 30 day average.

We set the value of $\sigma$ to a fraction, or coefficient of variation (Cv), of the observational value. For the plentiful daily
data values and especially those which can approach or straddle zero we also defined a minimum (min) $\sigma$ to ensure that the
uncertainty never dropped below a minimum (see Table 5). The values chosen reflect our best estimate of the uncertainty in
the random error, often in the absence of very reliable information to quantify this. Our choices however, are informed by
knowledge of how the data were collected and also by literature (Table 2).

### 2.4.5    Sampling the posterior

Since BASFOR is a numerical model, no analytical solution to calculate P($\theta$|D) is possible. We therefore chose a Monte Carlo
sampling based method to approximate the posterior. An efficient sampling method is Markov Chain Monte Carlo (MCMC)
and the simplest algorithm was proposed by Metropolis et al. (1953). Further details of our approach to Bayesian calibration
at a single forest site can be found in Van Oijen et al. (2005).

### 2.4.6    Assessing convergence in the MCMC chain

Convergence was assessed by running two MCMC chains and testing to see whether the within chain variation was larger than
the between chain variation. This was done using the standard Gelman-Rubin statistic. The starting points of the two chains
were chosen to be 20% and 80% of the total range of the prior respectively. Convergence was evaluated after running MCMC
chains of length 100000.

### 2.5    Three kinds of Bayesian Calibration (BC)

In this study we made three kinds of calibration.

### 2.5.1    The default multi-site calibrations including all observational data

In sections (3) and (4) we present results the control or default calibrations where pine, spruce and deciduous forest sites are
calibrated at the same time and all the observational data are included in the calibration.





### 2.5.2 Multi-site calibration with selected observational data

In section (5) we present results from calibrations where pine, spruce and deciduous forest sites are calibrated at the same time with different sets of observational data included as defined below.

1. In this default case, as already discussed in the previous subsection, all the observational data are included in the calibration: Labelled "All".

2. Only the low frequency data (aboveground vegetative carbon, carbon in the roots, carbon in the leaf litter, soil organic carbon, diameter at breast height, height and leaf area index) are included in the calibration: Labelled "LowFreq".

3. Only soil emission data (soil respiration, NO and N2O emissions) are included in the calibration: Labelled "SoilEm".

4. Only 30 day averaged evapotranspiration flux data is included in the calibration: Labelled "ET".

5. Only 30 day averaged net ecosystem exchange flux data is included in the calibration: Labelled "NEE".

6. Only 30 day averaged soil water content data is included in the calibration: Labelled "WC".

### 2.5.3 Site-specific BC including all the observational data

In section (6) we compare results from calibrations for pine forests for the default multi-site calibrations with all the data included: Labelled "pooled" (and identical to the default calibration labelled "All" above) against calibrations where each pine forest site is calibrated separately: Labelled "unpooled".

### 2.6 Quantifying the influence of the calibration on model-data differences and uncertainty

We used two main measures to quantify how the model had changed after calibration. The first quantified how the model-data difference had changed as a result of the calibration. While neither model nor data are without error, a closer match after calibration is generally indicative of an improvement in model performance. Model-data differences were quantified using the root mean square (RMS) difference of the observations and the model run with the maximum a-posteriori (MAP) parameter vector, since this was the most probable model prediction given the data, and prior mode parameter vectors as this represents the most probable parameter vector prior to the comparison against data. The ratio of MAP to prior mode RMS deviations was used to quantify how model-data differences had changed after calibration. Values in this ratio greater than one represent a smaller RMS deviation in the prior than posterior and vice versa.

We also quantified how uncertainty in the model outputs had changed after the calibration. Van Oijen (2017) states that uncertainty is having incomplete knowledge of a quantity and can always be represented by a probability distribution. Therefore, To quantify uncertainty after calibration we sampled from the posterior MCMC chain after removing the burn-in. This was done by sampling 1000 points equi-spaced along the MCMC chain. Since each sampling of the chain was spaced sufficiently far apart this was in effect a random sample. BASFOR was then rerun with parameter vectors from the sample and model





output was generated. The uncertainty in model output was summarised by calculating the 5th and 95th quantiles. For comparison, the prior uncertainty was also quantified by taking a random sample of 1000 from the marginal Beta prior distributions that were used in the calibration. Once BASFOR was rerun the with the sample the 5th and 95th quantiles were calculated to summarise the uncertainty, as above for the posterior. To quantify how uncertainty had changed, we analysed how the width

of the probability distribution of the model outputs changed from prior to posterior. The change was quantified by forming the ratio of posterior interval 95th - 5th quantiles over prior 95th - 5th quantiles. A ratio smaller than one indicates a narrower posterior than prior and hence a reduction in uncertainty and vice versa.

## 3 An example of how BC influenced the time-series output of BASFOR.

Before going on to present time-averaged summaries for all forest sites, we first present examples at two sites, one coniferous

(Hyytiälä) and one deciduous (Griffin), of how parameter calibration influenced the time-series output of BASFOR.

In Figs 1 and 2 are shown examples of time-series output of BASFOR before and after calibration for outputs where there were a lot of data available (Table 3) namely Net Ecosystem Exchange (NEE), evapotranspiration (ET) and soil water content (WC) and also much more sparse "low frequency" data (above-ground carbon (CLBS), soil organic carbon (CSOM), diameter at breast height (DBH), height (H), and leaf area index (LAI)).

### 15 3.1 High frequency data (carbon fluxes, water fluxes and soil water content)

Fig. 1 shows that after calibration the model output is closer to the observations for NEE, GPP and WC. Although in the case of the later there remain significant differences in phasing and variability between the model and observations. For evapotranspiration the posterior remained close to the prior. Model output uncertainty is reduced from prior to posterior.

### 3.2 Low frequency (ancillary) data

Fig. 2 shows that the calibration brought the model closer to observations for the above-ground carbon pools (CLBS), height, DBH and LAI whereas the model is further away from the single soil carbon measurement after calibration. As for the fluxes uncertainty in the model outputs are reduced after calibration.

## 4 Changes in model outputs after multi-site calibration

Before considering how multi-site calibration differs from calibrating at each forest site separately and which observations

were most effective in the calibration, we first present a summary of the differences the multi-site calibration made to the model outputs.

We use the method of analysis reported in section (2.6). We analyse and summarise how model-data differences changed after calibration by comparing model output against data from the most likely parameter vector prior to calibration (the prior mode) and the most likely parameter vector after calibration (the maximum a posteriori (MAP)) using root mean square (RMS)





deviation. We evaluate and summarise how prior output uncertainty due to parameter uncertainty has changed by forming the ratio of posterior uncertainty over prior uncertainty averaged over the period that observations were available.

The results are summarised in grid form of model outputs versus sites where the model outputs presented are those for which we had calibration data (Figs 3 and 4). The exception is GPP which was not included as calibration data since it is essentially
a model derived quantity. Box and whisker plots which summarise further over sites and output variables can be found in the supplementary material.

### 4.1 The influence of calibration on model - observation RMS deviations.

We first consider which of the model outputs showed the largest change in RMS deviation from data after calibration.

Of the low frequency model outputs, aboveground vegetation (CLBS), roots (CR), DBH and LAI generally show the largest
reductions in model data differences after calibration. Whereas height (H) and soil organic matter carbon (CSOM) show the largest increases in model-data differences. Model-data differences in height were generally smallest prior to the calibration which partially helps to explain the lack of improvement after calibration.

For the low frequency variables, model outputs at sites NL-Spe and IT-Ren generally moved furthest away from the observations, relative to the prior mode, after BC, although IT-Ren was one of the sites closest to the observations prior to the BC.
Model outputs for forest sites SE-Nor, CZ-BK1, DE-Tha and RU-Fyo also moved further away from observations after calibration. The largest spread in the low frequency variable RMS ratios was for sites SE-Nor, FI-Hyy, DE-Tha, IT-Ren, DE-Hai and CZ-BK1 with the variable CSOM and sometimes H being the cause of the wide spread with other low frequency outputs becoming closer to the observations after calibration. Outputs for sites ES-ES1, SE-Nor, IT-Col and DE-Wet were furthest away from the observations before calibration.

For the high frequency variables, model outputs, soil water content (WC) was closest and GPP furthest away from observations before calibration. WC also showed the greatest relative reduction in RMS deviation after calibration whereas ET showed the greatest relative increase in RMS. In general, there was a more consistent relative reduction in RMS deviation after calibration than for the low frequency outputs which might be expected given the larger quantities of flux and WC data included in the calibration.

For the high frequency variables, the forest sites that showed the largest relative reduction in model-data differences after BC were FI-Hyy, DK-Sor and DE-Hai. Sites that showed least reduction or even increases in relative RMS deviation after BC, included NL-Loo, SE-SK2, ES-ES1, IT-Ren and RU-Fyo. Of these model outputs for NL-Loo was one of the closest sites to observations prior to BC. Model output at sites IT-SRo and UK-Gri were furthest from the observations prior to the calibration.

### 4.2 Changes in uncertainty after calibration

From Fig. 4 we see that uncertainty has generally been reduced as a result of the BC. The small number of exceptions are when the prior range of a model output was restricted because the values were close to zero and negative values were not possible. For example, prior values of LAI were close to zero for a number of sites prior to the calibrationx.



There is also a much stronger relationship between the number of observations included in the calibration and the reduction in uncertainty from prior to posterior than was true for RMS deviation changes after calibration.

It is also noted that uncertainty reduction was less where the variance in time of the model output increased significantly from prior to posterior. This was true for example for the output GPP at sites SE-Sk2 and SE-Nor.

## 5  How different sets of observations influence the calibration of BASFOR

Since we have available a rich dataset encompassing many aspects of the forest ecosystem we evaluate how the inclusion of different sets of observations influences the calibration of BASFOR. For each of pine, spruce and deciduous forests six calibrations described in section (2.5.2) are run.

The results presented below are averaged across all the forest sites and hence also across pine, spruce and deciduous forests.

As previously, the results focus on how the model comparison against observations and how uncertainty has changed from prior to posterior (section 2.6).

### 5.1  Model comparison against observations.

Shown in Figs. 5 and 6 is the ratio of RMS deviation against observations of BASFOR run with the MAP parameters over the prior mode parameters for all six calibrations.

### 5.1.1  Comparison against high frequency data.

As might be expected, the largest improvement in the comparison between model and observations occurs when the calibration includes the data for the variable being compared. For example, RMS deviations decrease more for NEE when NEE is included in the calibration (Fig. 5). Model derived GPP was not included in the calibration so in this case the largest decreases are found when all the calibration data are included although as might be expected inclusion of NEE is the next most important dataset for decreasing model and data differences.

Not all the calibrations improved the comparison against observations. Indeed, the "ET" calibration increases model differences against the NEE and GPP data and similarly the "NEE" calibration slightly increases differences with the ET data. The calibration including just the low frequency data ("LowFreq") increases differences with the NEE and particularly ET data but slightly improves the comparison with the GPP and soil WC data.

### 5.1.2  Comparison against low frequency data.

Unlike the comparison against high frequency data, inclusion of the low frequency data on its own does not always lead to the greatest improvement in the comparison against observations. Model outputs with the lowest average number of observations included in the calibration (Table 3) (CR, CLITT, CSOM) are less likely to show greatest improvement with the "LowFreq" calibration, whereas those with the highest number (LAI, H) tends to have the largest improvement (Fig. 6). Indeed, the model comparison against CSOM observations often gets worse after calibration regardless of whether ancillary data were included.



There are generally no more than a few observations of CSOM and the data have a high uncertainty (Table 5), so CSOM data have a relatively low influence in the calibration.

There is a greater spread in changes in RMS deviation versus the prior than we saw for the more plentiful flux data (Table 3), reported in the previous section. This is especially apparent for height, where only those calibrations that included height

data improved the comparison against observations relative to the prior. It was noted in section 4.1 (Fig 3) that H output from a run with the prior mode was closer to the data than for many other outputs making it harder to make improvements relative to the prior for height.

The "ET" calibration is most likely to increase differences with observations after calibration. This is interesting given what was noted against NEE observations previously and that many of the ancillary variables "measure" in some way carbon se-

questration. It is also worth noting however, that the "NEE" calibration also leads to increased differences (H, CSOM) or small improvements versus the prior (CLBS, DBH) although in some cases (CR, LAI, CLITT) there is a more significant improvement. Indeed, consulting individual plots of model against data for CLBS and height (not shown) the above ground carbon is often greater for the MAP run than for the prior mode and observations. For the "ET" calibration the situation is frequently reversed with the posterior aboveground vegetation being lower after calibration than before and against observations.

A similar picture also emerges for the height data comparison. This may suggest that the "NEE" and "ET" only calibrations have a tendency to lead to calibrations with relatively higher/lower carbon accumulation than would be expected from the observations.

## 5.2  Influence of BC on model output uncertainty

Shown in Figs. 7 and 8 is the ratio of 5th to 95th quantile range of the posterior over the prior for the six calibrations. This is a

measure of how uncertainty has changed in the outputs of BASFOR as a result of the calibration.

### 5.2.1  Changes in uncertainty after calibration for high frequency variables.

For the carbon fluxes, water fluxes and soil water content the uncertainty is reduced in all six calibrations. The reduction is largest when all the data are included in the calibration, followed by the calibration of the same data type as the output. For GPP the largest reduction after all data is for "NEE" which might be expected. The "ET" calibration is next most important

for NEE, GPP and WC with the "NEE" being the next most important for evapotranspiration. The "LowFreq" calibration was consistently the fourth most important regardless of it including fewer data than "NEE" or "WC". For WC, "NEE" was of less importance although, the least important calibration data was "SoilEm" for WC and ET. Consistent with this "SoilEm" and "WC" calibrations had the smallest impact on uncertainty for NEE and GPP.

### 5.2.2  Changes in uncertainty after calibration for low frequency variables.

As for the low frequency variables, the uncertainty is generally reduced after calibration. The calibration which includes all the observations "All" followed by "LowFreq" are generally the most effective in reducing uncertainty though not for LAI and




CLITT. The "ET" calibration is the most important for reducing uncertainty for LAI and CLITT although, referring to Fig.(6) this calibration also leads to the largest increase in RMS deviation for those model outputs. Indeed, more generally, while "SoilEM" and "ET" calibrations are relatively important for reducing uncertainty in plant and litter carbon (CLBS, CR and CLITT) this does not lead to reductions in RMS differences (Fig. 6). The calibration "NEE" is more important for reducing

uncertainty for outputs with large stores of carbon (CLBS and CSOM) but from Fig.(6) this is not accompanied by reductions in RMS differences. "WC" is generally the least effective in reducing uncertainty for the ancillary outputs which is consistent with what was noted previously for NEE and GPP.

## 6    A comparison of unpooled and pooled calibrations for pine forests.

As described in section 2.5.3, in this section we return to calibrations where all the observations are included to compare the

influence of making separate calibrations at each pine forest site (unpooled calibrations) with a calibration where all pine forest sites were calibrated together (pooled calibration).

### 6.1    Influence of pooled versus unpooled calibration on model - data RMS deviations.

Shown in Fig. 9 are results for outputs where more data were included in the calibration (Table 3). Unpooled calibrations show greater improvement than pooled calibrations for NEE. Outputs GPP and ET show no great difference with WC showing a

slightly greater improvement for the pooled calibration.

For the outputs which generally have fewer observations included in the calibration (Fig. 10 and Table 3) CLBS, H and LAI have greater improvements for the unpooled calibration whereas CR, CLITT and CSOM show greater improvements for pooled. Consulting Table 3 this suggests that outputs with very few observations tend to benefit more from the pooled calibrations where observations from all the pine sites can influence the calibration.

Examining the same results now disaggregated by site we see that for the outputs with lots of data (NEE, ET and WC) (Fig. 11) only one site, FI-Sod, shows greater improvement for the pooled calibration whereas for the outputs of low frequency data (Fig. 12) three sites FI-Sod, IT-SRo and NL-Spe have greater improvement. For the site FI-Sod, the pooling of information from the other pine sites affects the extent to which low CLBS data values are influential in the pooled calibration. This is also beneficial for the fit to data of other outputs at FI-Sod (Fig. 13). For forest sites IT-SRo and NL-Spe, low quantities of LAI,

CLBS and CSOM data are supported by the influence of observations at other sites to give greater improvements for those outputs in the pooled calibration (Fig. 13). In consequence, there is a slight detrimental effect for outputs with more numerous data in the pooled calibration at NL-Spe and IT-SRo (Figs. 11 and 13). Sites ES-ES1 and NL-Loo show particularly large improvements of unpooled over pooled calibrations (Fig. 12.

### 6.2    Influence of pooled versus unpooled calibration on prior to posterior uncertainty changes.

Comparing the change in uncertainty from prior to posterior for the pooled and unpooled calibrations (Figs. 14 and 15) there is in general a greater reduction in uncertainty for the pooled calibration. The exceptions are WC and CLITT where there is little





difference in the reduction in uncertainty. This difference between pooled and unpooled is generally larger for the low frequency outputs which had fewer calibration data (Table 3). There is also a greater range between sites, of changes in uncertainty for the unpooled than for the pooled calibration. This difference in range between pooled and unpooled is particularly evident for H and DBH.

## 7  Discussion

### 7.1  How does a rich observational dataset benefit forest process-based model calibration?

Access to good quality observations that represent more than just a few of the ecosystem variables under study, for calibrating PBMs, remains a significant challenge. While there are now increasing numbers of studies where ecosystem and even forest PBMs are calibrated at multiple sites, most studies to date have had access to data limited to just one or two aspects of the

ecosystem. For example, Lehuger et al. (2009) calibrated against $N_2O$ emissions, Van Oijen et al. (2013) used height and DBH data and Molina-Herrera et al. (2015) had only carbon flux data. In this study, we had access to observations, for calibration, on a significantly greater number of European forest ecosystem variables and at more forest sites than previous studies.

The results demonstrate that while most model outputs had a closer fit to data after calibration this was not always the case. Some model outputs, for example height, were relatively close to the data prior to the calibration making further improvements

difficult. For other quantities, such as soil organic matter, there were very few observations and the uncertainty was set high, reflecting a lack of confidence in the accuracy of the data, hence, this data had relatively little impact on the calibration. In general, there were more consistent reductions in model-data differences where a greater number of calibration data, such as NEE or soil water content observations, were available. In contrast, evapotranspiration, for which there were also a relatively large number of data points, had increased RMS differences after calibration, at some sites. Such increases in a-posteriori

model-data differences could suggest inconsistencies between the data and the model, so that improvements in the model representation of some parts of the ecosystem come at the expense of others. We will return to this subject when discussing the results of calibrations with different datasets below.

There was considerable variation in the extent to which model simulations improved after calibration at the different forest sites. This would indicate either a variation in the accuracy of forest data from different sites or a variation in the ability of the

model to represent some of the sites which may be atypical of European forests in general or both. We will discuss this further in the next section when comparing pooled and unpooled calibrations.

While it is often of primary concern whether the model fit to data improved after calibration, it is difficult to assess the value of model predictions without making an assessment of uncertainty. Since we use mathematical probabilities to combine models and data, we can quantify uncertainty propagated from the parameters through to model predictions that have a precise meaning

mathematically, aiding interpretation. There was a consistent reduction in uncertainty after calibration with a strong relationship between the number of data-points included in the calibration and uncertainty reduction. Uncertainty only increased where the range of prior uncertainty was constrained, due to prior model values being close to zero for a field such as LAI that cannot be negative.




### 7.1.1 How effective are different observational datasets for reducing model-data differences in BC?

An important question is how effective different observed datasets are for decreasing model-data differences and reducing parameter and therefore model prediction uncertainty. This study is well placed to look at this question since the calibration dataset used in this study is particularly rich, both in terms of number of different forest sites included and also the breadth of

the ecosystem represented by the data.

A number of findings can be taken from the results. Most obviously, reducing model-data differences is most effective when the data used in the comparison are also included in the calibration. Secondly, the single dataset calibrations highlighted where there were inconsistencies either between the different observational datasets or between the observations and the model or both (subsequently referred to in the text as just "inconsistencies"). The single dataset calibrations also shed light on how these

inconsistencies had influenced the calibration with all the observations included. For example, a calibration including just NEE observations increased model-data evapotranspiration differences and vice versa. The "NEE" calibrations had a tendency to give higher aboveground carbon accumulations than were expected from the observations whereas the "ET" calibration had a tendency to give lower aboveground carbon accumulations than those expected from observations. This indicates that there could be inconsistencies between the observations of NEE, evapotranspiration and aboveground-carbon and the model. These

inconsistencies hampered reductions in model-data differences in these outputs when all the calibrations data were included. The results from the calibrations cannot help to disentangle the source of the issue but they can help to diagnose where there may be conflicts between the model and the data and also between datasets. Where the calibrations showed smaller inconsistencies, such as for the NEE and soil water content only calibrations and root carbon, DBH and LAI data, there were greater reductions in model-data differences for root carbon, DBH and LAI in the calibration with all the data included. Similarly, while NEE

and evapotranspiration (ET) had approximately equal numbers of observations and accuracy, NEE data were more influential than ET data in the calibration when all the data were included. The evapotranspiration only calibration had larger model-data differences with the carbon stock data than the NEE only calibration. This may help to explain why the model was generally closer to NEE data but further from the evapotranspiration observations at some sites after calibration, as already noted. In general, calibration data was less effective in reducing model-data differences when there were larger inconsistencies present.

Further, the results indicated a relationship between the size of the inconsistencies and the quantity of calibration data required to influence the calibration, when all the observations were present. Where calibrations that included just one data-type suggested that there was a relatively large increase in model-data differences after calibration (for example, NEE and evapotranspiration only calibrations against height observations and the NEE only calibration and soil organic carbon (CSOM) observations) this would suggest a larger inconsistency. For larger inconsistencies the data (height or CSOM) had to have

relatively greater "weight" in the likelihood function, either through greater accuracy or a larger number of observations, to influence the calibration when all the data are included in the calibration. While there was a sufficient quantity of height data, included in the calibration, to result in reduced model-data differences for height when all the calibration data was included, the few CSOM observations included were largely ignored by the calibration and model-data differences for CSOM increased after calibration.





These results highlight the danger that if only one or two aspects of the ecosystem are constrained by observations in the calibration then, since models are imperfect, the fit can be good for the parts of the ecosystem represented by the data but at the expense of other aspects of the ecosystem, where the fit can worsen. Therefore, a calibration including data that represents a greater variety of ecosystem variables, is a more stringent test of an ecosystem PBM, since the model has to stay consistent

with the greater diversity of the observed ecosystem, for there to be a good fit to the observations after calibration.

Although there were significantly fewer carbon stock than flux data available for the calibrations, these more sparse observations were generally more influential for reducing model-data differences in carbon stocks, than the much more plentiful carbon and water flux data.

### 7.1.2 How effective are different observational datasets for reducing uncertainty in BC?

As would be expected, the greatest reduction in uncertainty was generally when all the observations were included in the calibration. Uncertainty in model output is reduced the most when observed data representing that output is included in the calibration.

Uncertainty was reduced more where there was a strong relationship between the model output and the data included in the calibration. For example, the second largest reduction in uncertainty in GPP and also larger reductions in carbon pools above

and belowground was for the calibration that included just NEE observations. Indeed, the relatively sparse low frequency data had a disproportionately large influence on reducing uncertainty in the carbon stocks but also in the carbon fluxes relative to the much more numerous soil water content data. Further, large reductions in uncertainty also occurred when the observations included in the calibration had a tendency to increase model-data differences. For the example, the inclusion of ET/NEE observations caused the third largest reduction in uncertainty in NEE/ET. Therefore, a large reduction of uncertainty in the calibration

does not imply that the model has become much more accurate. While these reduced uncertainties are mathematically correct, if we assume that the model and data do not have structural errors and systematic biases, the small a-posteriori uncertainties are not useful if the models are to be used for predicting the future of European forests, with appropriate uncertainties quantified. To improve the utility of the uncertainties that are quantified, we need to represent uncertainty about model structural error and systematic observational biases in the Bayesian calibration. Indeed, diagnostic information about where models and data

are in conflict, uncovered by the calibrations in this study, could help in the construction of improved likelihood functions that include representations of model structural errors and data biases.

### 7.2 Are separate calibrations at forest sites more effective for improving model fit to data and reducing uncertainty than multi-site calibration?

Previous studies (Reinds et al., 2008; Lehuger et al., 2009; Van Oijen et al., 2013; Van Oijen and Höglind, 2015; Molina-Herrera

et al., 2015) have recognised that calibrating model parameters at multiple sites gives protection against a model losing general applicability through calibration for a very atypical site. However, there is disagreement about whether model performance is improved by calibrating separately at each forest site (Reinds et al., 2008; Lehuger et al., 2009; Molina-Herrera et al., 2015) or not (Van Oijen et al., 2013). Since previous forest studies had access to more limited calibration data at multiple sites than here,



we revisited the question of whether pooled or unpooled calibrations were more beneficial for reducing model-data differences and uncertainty, aiming to provide a more definitive answer than was possible before.

We found a relationship between the quantity of data included in the calibration and whether a pooled or unpooled calibration was more beneficial for reducing model-data differences. In general, where more data were included, the unpooled calibration

reduces RMS difference with data more than the pooled calibration whereas, if very few observations are available at a forest site, the pooling of observations from other forest sites was often more beneficial.

This is consistent with what was found previously because studies with relatively larger quantities of calibration data (Reinds et al., 2008; Lehuger et al., 2009; Molina-Herrera et al., 2015) reported a benefit for unpooled calibrations whereas, those with fewer data (Van Oijen et al., 2013) did not find an improvement.

Pooled calibrations were more beneficial for sparse observations because the influence of low data quantities in the calibration increases when pooled with other sites. Therefore, pooled calibrations can help when datasets have large imbalances in the quantity of data available. Since it is likely that obtaining observations for some parts of the forest ecosystem, for example eddy-fluxes, will remain easier than for others, such as belowground carbon amounts, some pooling in calibrations will remain important.

Calibrating forest models with observations at many sites is a good test of the generality of the processes represented in the model and can protect against decreasing the generality of the model predictions by calibrating to a very unrepresentative site. We found that for some sites (ES-ES1 and NL-Loo) the difference in the reduction in model-data deviations between the unpooled and pooled calibrations was particularly large. Since these sites benefited more from being calibrated separately, this may suggest that they are more atypical. Indeed, El Saler (ES-ES1) is a mixed grass-tree ecosystem and hence is less likely to

be represented well by a tree only model such as BASFOR.

We found that the pooled calibration led to a greater reduction in uncertainty than the unpooled calibration. This is expected given the greater quantity of data included in the pooled calibration. Differences in uncertainty reduction for pooled and unpooled was greater for outputs with fewer observations. There was also a greater consistency in uncertainty reduction across sites for the pooled calibration. This greater consistency was most marked for height and DBH suggesting that these

observations were the most consistent across forest sites.

For parts of the ecosystem that had higher data quantities available, the calibration showed a slight detriment in the pooled versus the unpooled calibration. This could suggest a conflict in the data or model structural deficiency or both. Indeed, with imperfect models and data and with significant variations between forest sites it is likely that parameters of BASFOR are not totally generic across sites. The best compromise would be to allow a variation in some model parameters across sites but

yet retain the pooling of information, especially where only a few observations are available. As recommended in Lehuger et al. (2009), a solution that allows parameters to vary but constrains the variation under a joint probability distribution, thus retaining the strengths of the pooled calibrations, is provided by hierarchical Bayesian calibration (Ogle and Barber, 2008). The evidence from this study would suggest that such an approach could be beneficial for the calibration of the parameters of process-based models for European forests.



# 8 Conclusions

## 8.1 Which observational datasets were most effective in reducing uncertainty in model predictions and model-data differences?

- When more calibration data were available for an ecosystem variable there were larger reductions in uncertainty and also more consistent reductions in model-data differences for that ecosystem variable after calibration.

- Calibrations including data that represents a greater variety of ecosystem variables, is a more stringent test of an ecosystem PBM, since the model has to stay consistent with a greater diversity of the observed ecosystem, for there to be a good fit to the observations after calibration. In other words, calibrations which include data from just a few aspects of the ecosystem can be problematic since improved model fits for the parts of the ecosystem represented by the data can be at the expense of other parts of the ecosystem where the model-data fit can worsen.

- The single dataset calibrations diagnosed where inclusion of data representing particular ecosystem variables led to significant increases or decreases in model-data differences in other ecosystem variables after calibration, suggesting a strong relationship between the underlying processes as represented in the model.

- Where the single dataset calibrations led to an increase in model-data differences in other ecosystem variables after calibration this could be due to systematic errors in the data or structural errors in the model or both. These errors hampered the reduction in model-data differences in the calibration with all the observations present.

- Where the single-dataset calibrations diagnosed a larger increase in model-data differences in other ecosystem variables after calibration a larger quantity of the other ecosystem data was needed in the calibration to reduce model-data differences.

- Where a stronger relationship between ecosystem variables was found there were also larger reductions in uncertainty after calibration.

- For some ecosystem variables uncertainty reduced after calibration but model-data differences increased. This would suggest that there were deficiencies in the model or systematic errors in the data or both.

- Sparse plant and soil stock observations were more important for reducing model-data differences and uncertainty in above and belowground carbon pools than more plentiful carbon and water flux data. Except for ecosystem variables where only very few uncertain observations were available.

- Uncertainty about model structural errors and systematic observational biases should be represented in Bayesian calibration so that uncertainty is not reduced inappropriately by calibration.



## 8.2 Is it more beneficial to calibrate forest sites separately or together ?

- While separate calibrations at each forest site generally reduced model-data differences more than calibrating at all the sites together, parts of the ecosystem that were very sparsely observed benefited more from multi-site calibration.

- Multi-site calibration helps a model to stay more generic by avoiding a calibration to a very unrepresentative forest site.

- Multi-site calibration led to larger and more consistent reductions in uncertainty than separate calibrations, especially for ecosystem variables with fewer observations.

- The best compromise is to allows variation in model parameters between different sites but to share information across sites to improve models. Such a compromise is possible with Bayesian hierarchical calibration.

*Competing interests.* The authors declare that they have no conflict of interest.

10  *Acknowledgements.* We would like to acknowledge the use of carbon, nitrogen and water flux monitoring data collected at long-term forest observation sites as part of the CarboEurope IP (http://www.carboeurope.org/) and NitroEurope IP (http://www.nitroeurope.eu) projects.




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





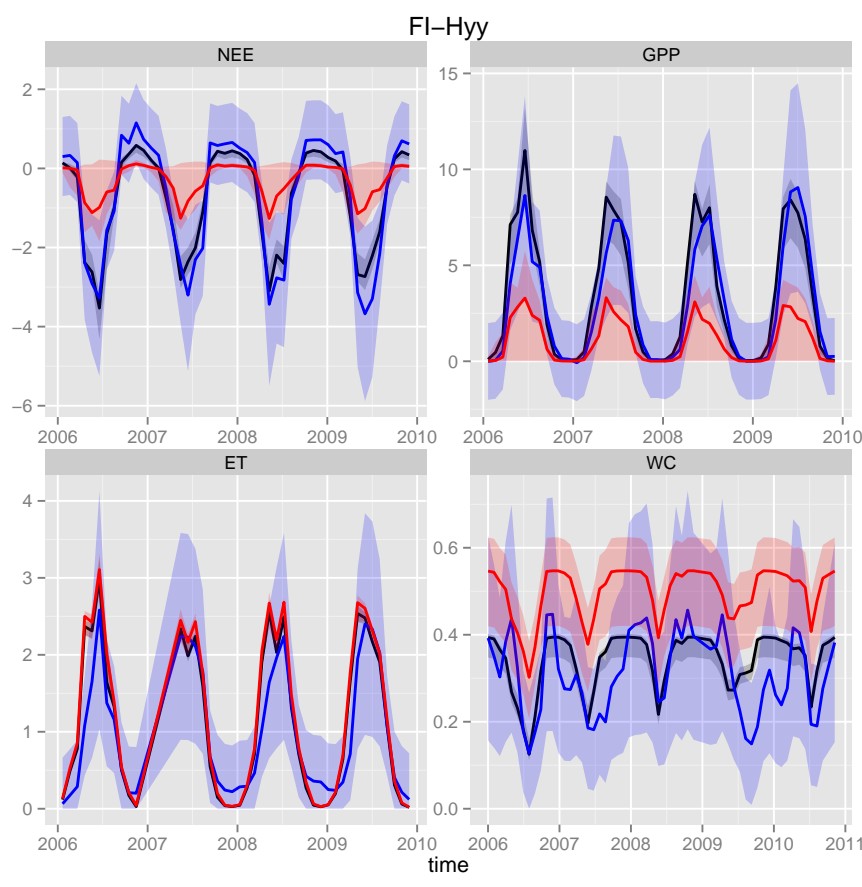

**Figure 1.** Timeseries of 30 day averaged carbon and water fluxes and soil water content for observations (blue line, with two standard deviations as used in the likelihood shaded blue) and the prior (the red line is the prior mode with the 5th to 95th quantile range shaded red) and posterior (the black line is the maximum a posteriori (MAP) with the 5th to 95th quantile range shaded) of BASFOR for Finnish site Hyytiälä.





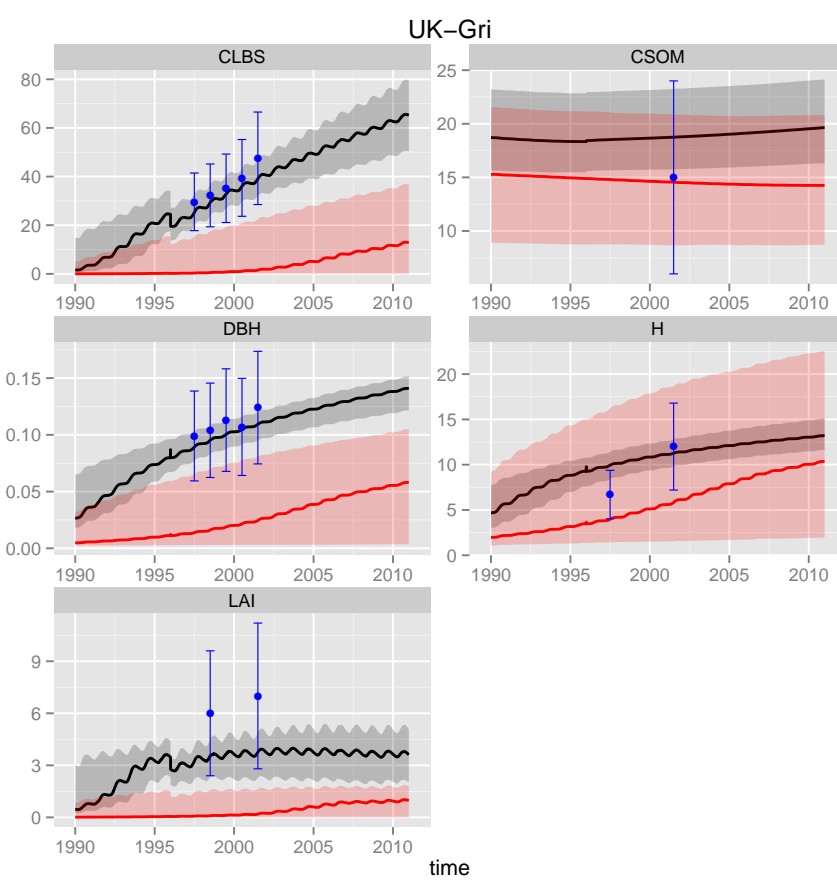

**Figure 2.** Timeseries of low frequency variables for observations (blue points, with two standard deviations as used in the likelihood marked) and the prior (the red line is the prior mode with the 5th to 95th quantile range shaded red) and posterior (the black line is the maximum a posteriori (MAP) with the 5th to 95th quantile range shaded) of BASFOR for UK site Griffin.



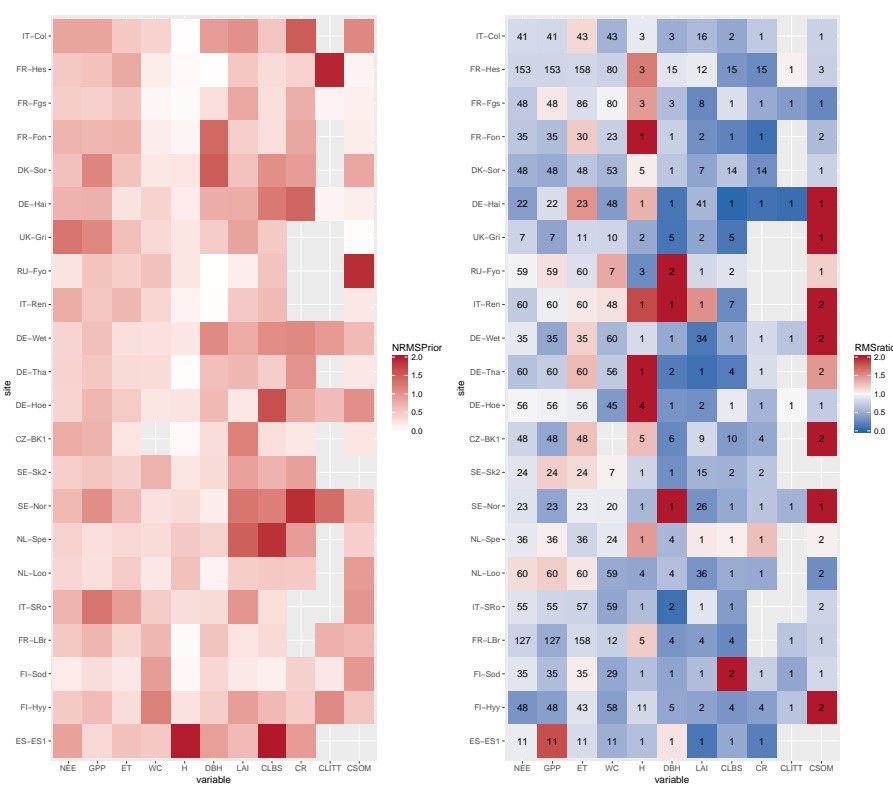

**Figure 3.** Left: Normalised RMS deviation against observations of BASFOR run with the prior mode parameter vector. Right: Ratio of RMS deviation against observations of BASFOR run with the MAP parameter vector over RMS deviation against observations of BASFOR run with the prior mode parameter vector. The numbers show the number of observations included in the calibration (minus GPP observations) and in the RMS deviation calculation. Pine sites are listed alphabetically at the bottom, spruce sites in the middle and deciduous sites at the top of the y axis.





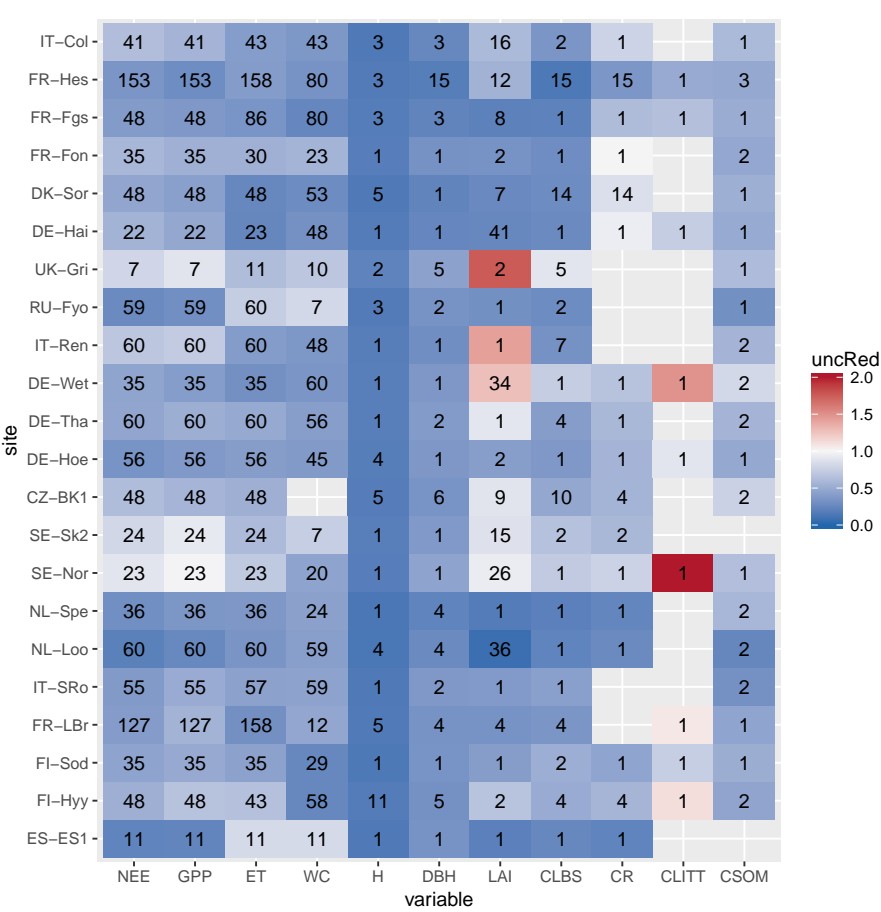

**Figure 4.** Ratio of the posterior to the prior of the range of the 5th to the 95th quantile. The numbers show the number of observations included in the calibration (minus GPP observations). Pine sites are listed alphabetically at the bottom, spruce sites in the middle and deciduous sites at the top of the y axis.





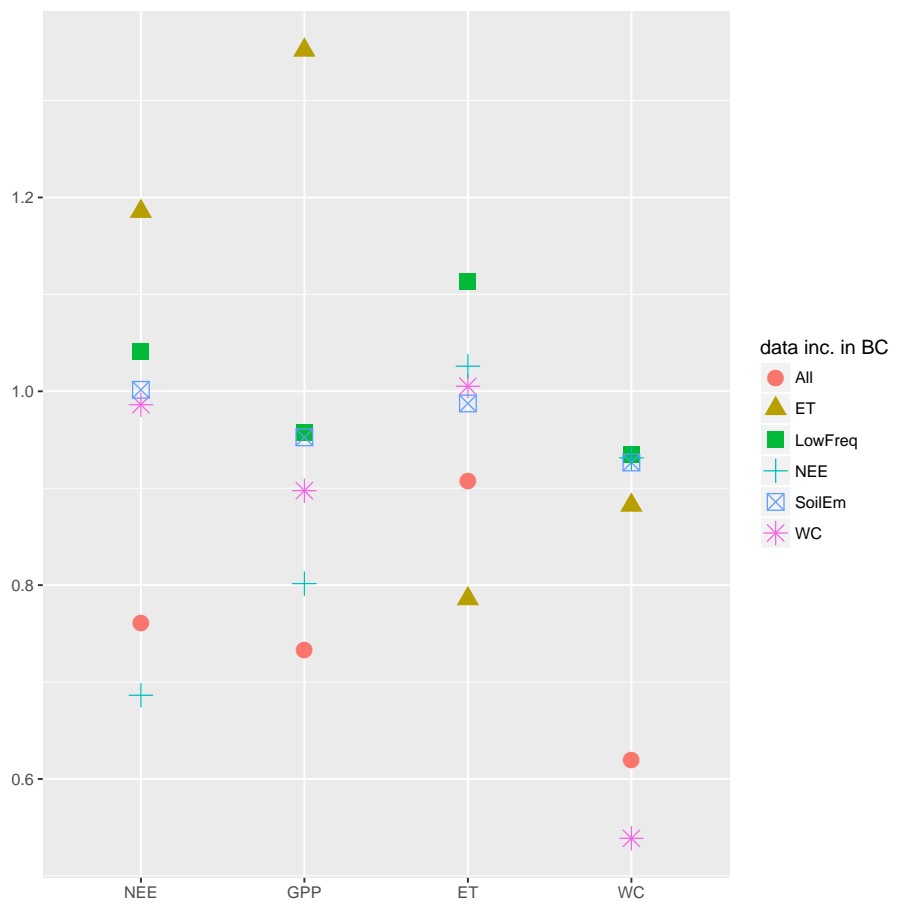

**Figure 5.** Ratio of posterior(MAP) over prior(mode) RMS deviation of model output and observations for six different calibrations including different sets of observations. See sections 2.5.2 and 5 for further details.





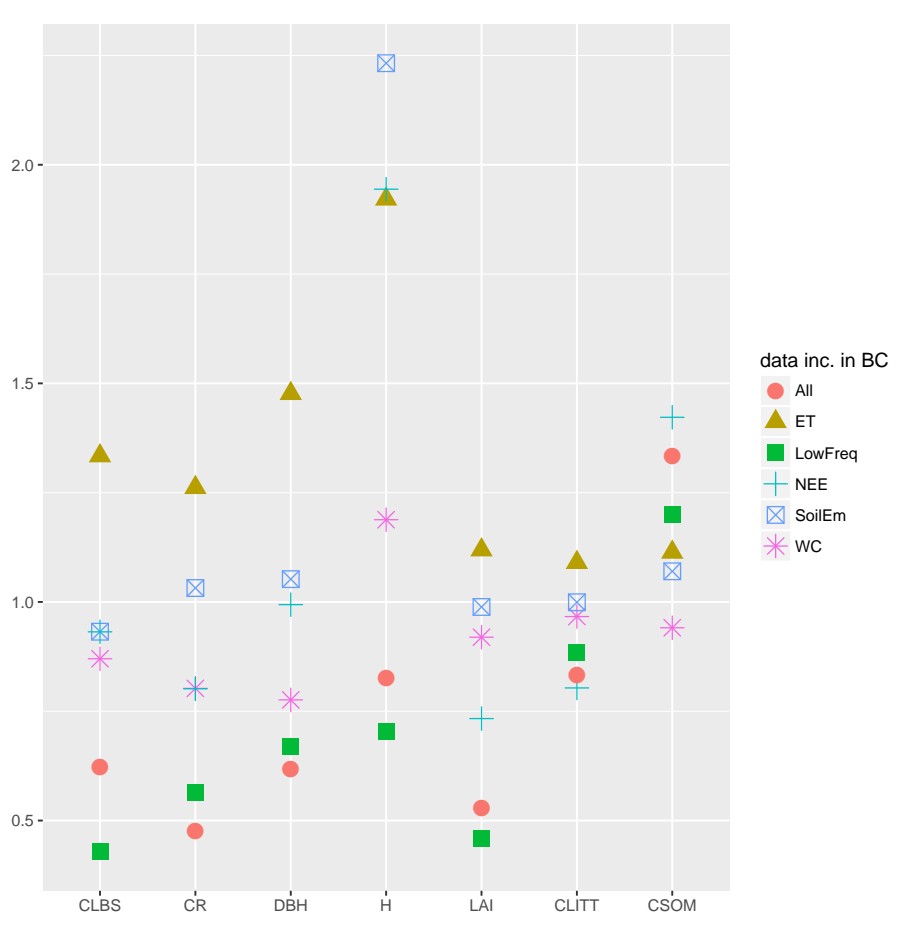

**Figure 6.** Ratio of posterior(MAP) over prior(mode) RMS deviation of model output and observations for six different calibrations including different sets of observations. See sections 2.5.2 and 5 for further details.



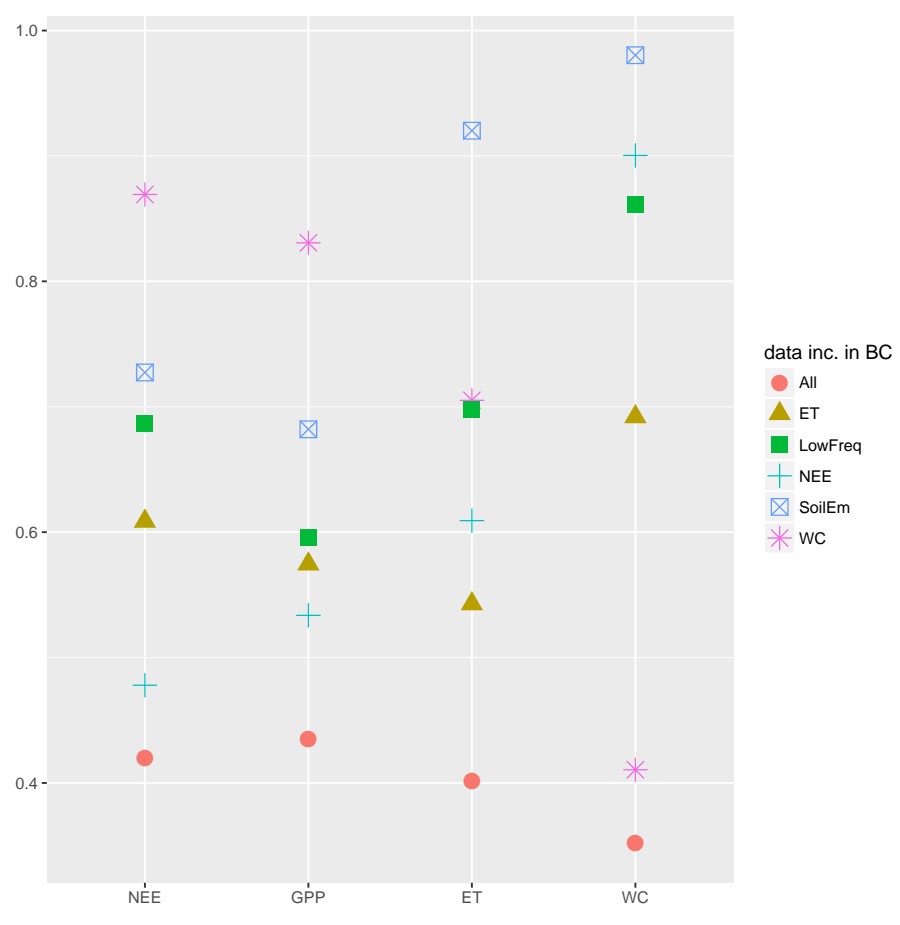

**Figure 7.** Shown is the ratio of 5th to 95th quantile range of the posterior over the prior for the six calibrations including different sets of observations. See sections 2.5.2 and 5 for further details.





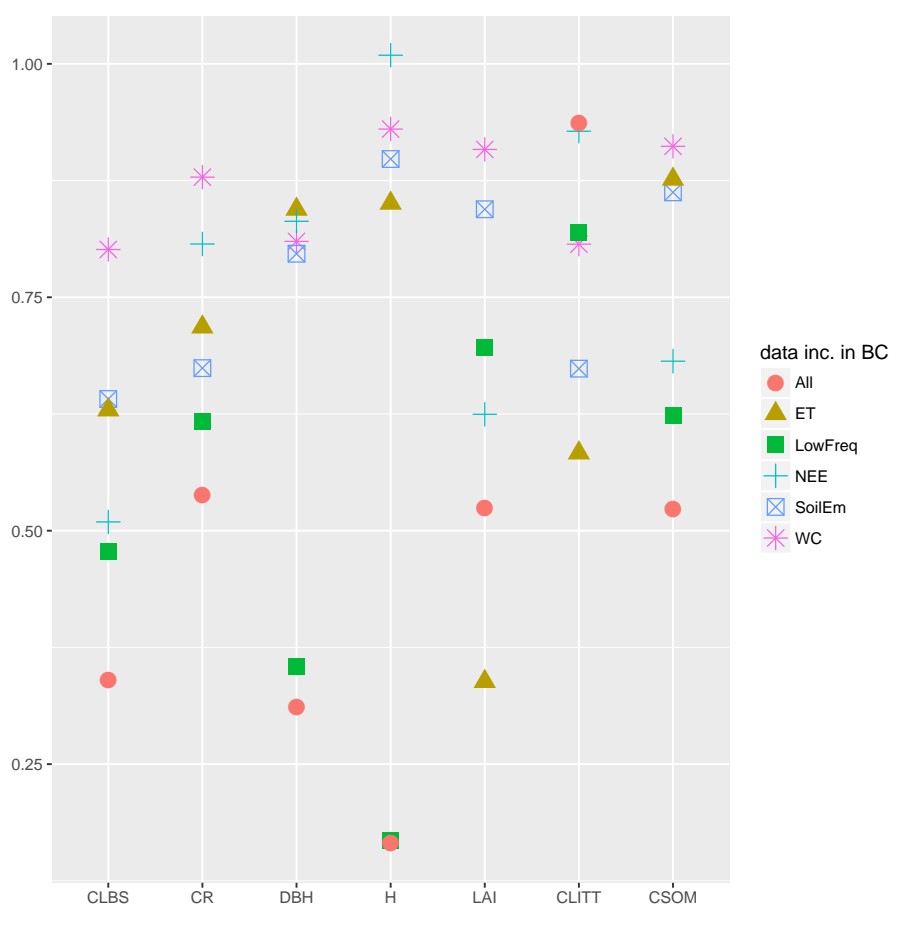

**Figure 8.** Shown is the ratio of 5th to 95th quantile range of the posterior over the prior for the six calibrations including different sets of observations. See sections 2.5.2 and 5 for further details.



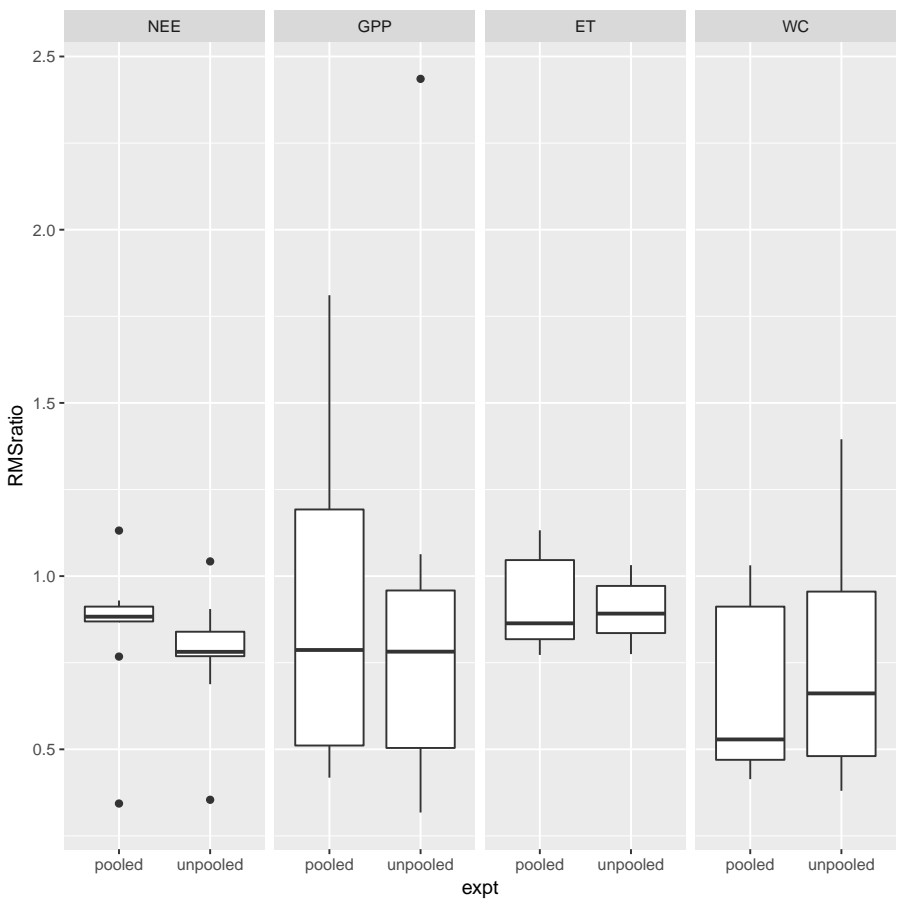

**Figure 9.** Ratio of posterior (MAP) to prior (mode) RMS deviations. Comparison of pooled and unpooled calibrations.





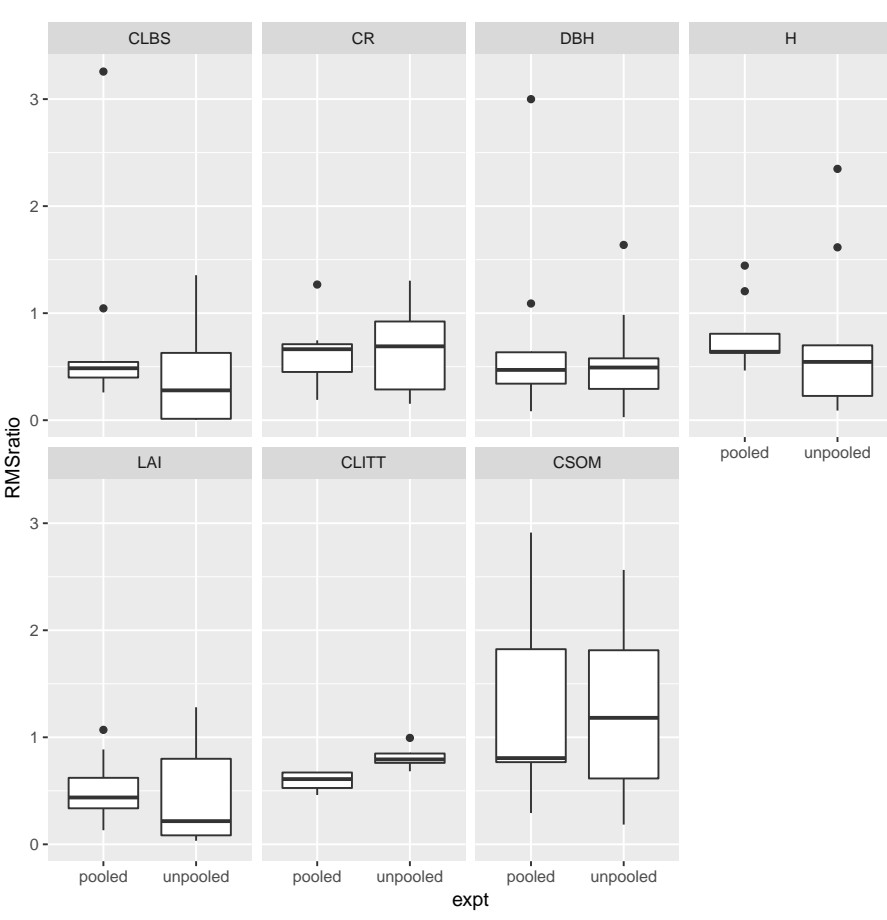

**Figure 10.** Ratio of posterior (MAP) to prior (mode) RMS deviations. Comparison of pooled and unpooled calibrations.





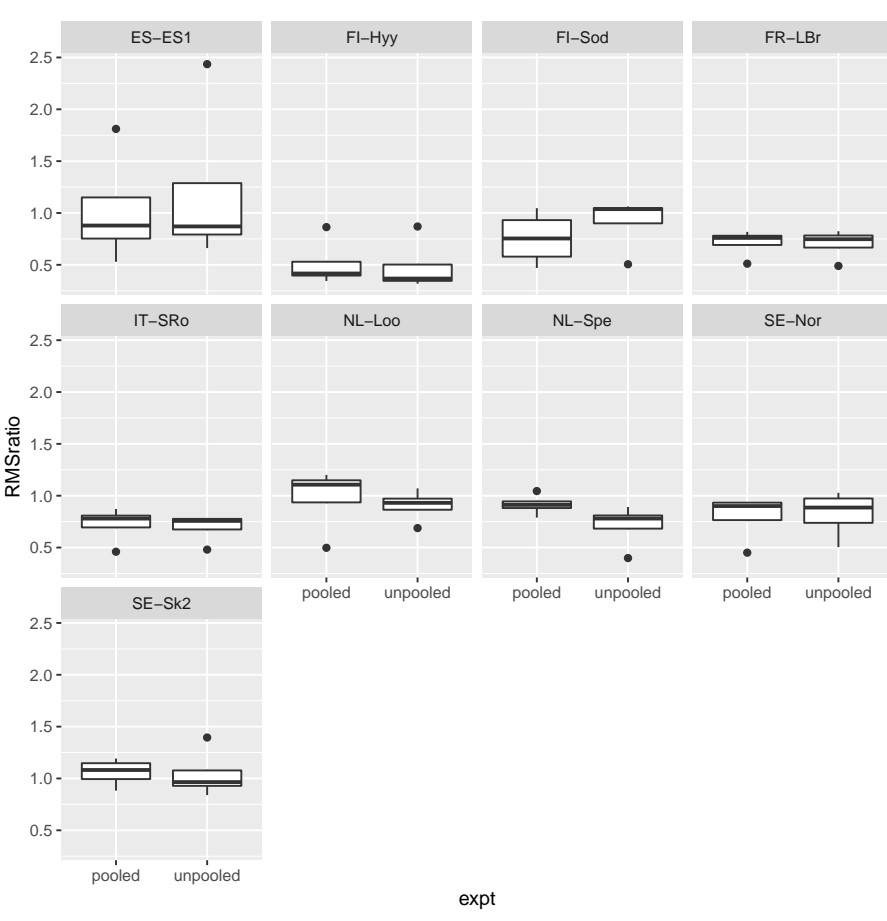

**Figure 11.** Ratio of posterior (MAP) to prior (mode) RMS deviations for NEE, GPP, ET and WC. Comparison of pooled and unpooled calibrations.





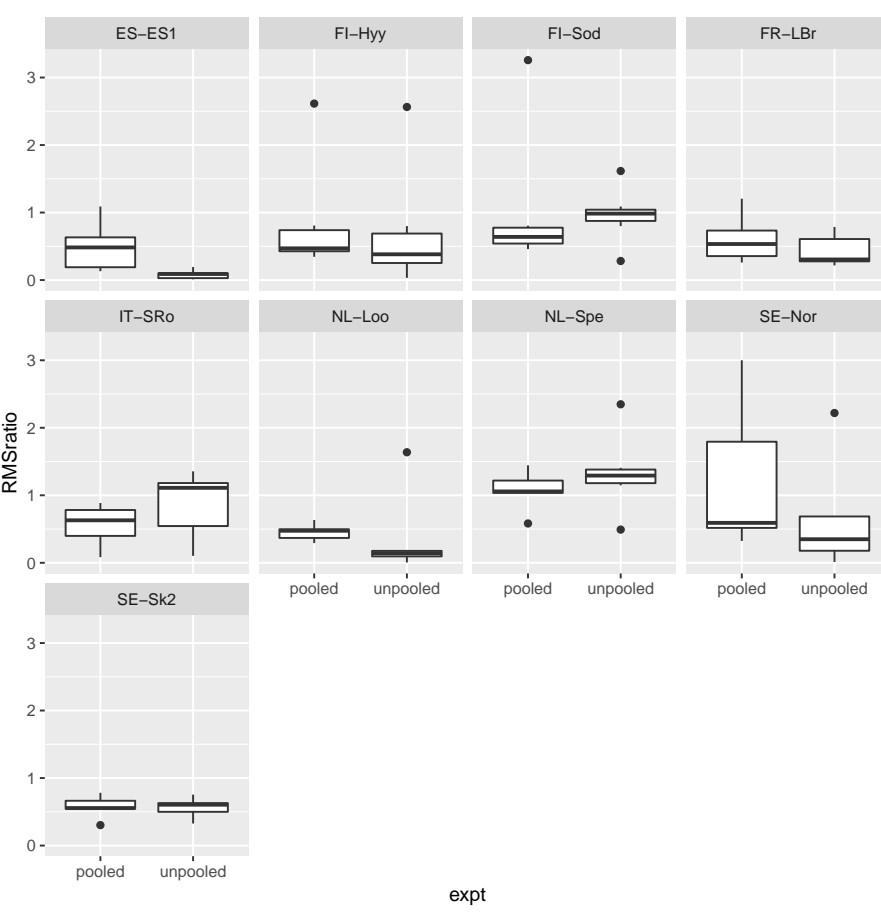

**Figure 12.** Ratio of posterior (MAP) to prior (mode) RMS deviations for the low frequency variables. Comparison of pooled and unpooled calibrations.



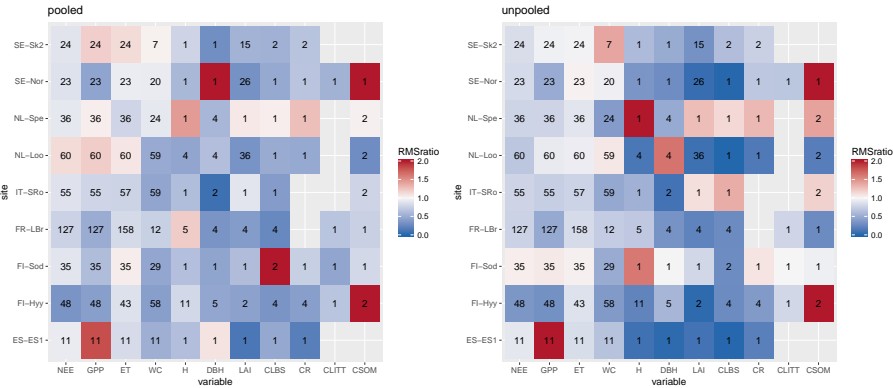

**Figure 13.** Ratio of RMS deviation against observations of BASFOR run with the MAP parameter vector over RMS deviation against observations of BASFOR run with the prior mode parameter vector. The numbers show the number of observations included in the calibration (minus GPP observations) and in the RMS deviation calculation.

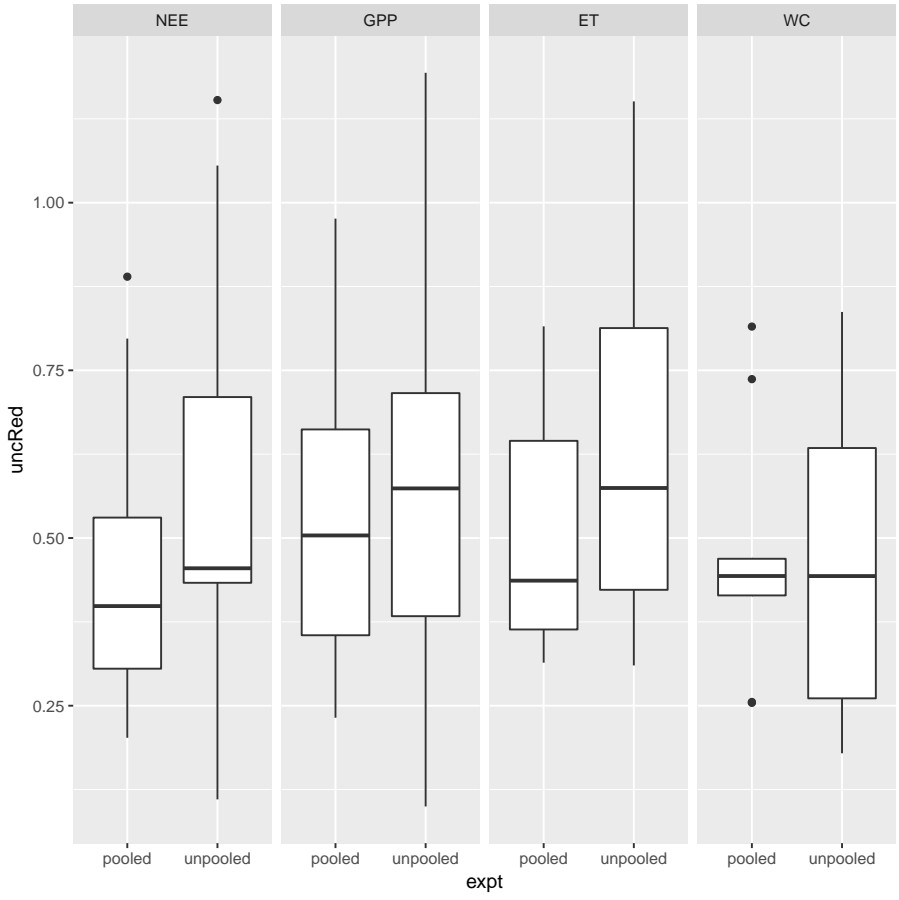

**Figure 14.** Ratio of posterior to prior 5 to 95 quantile range. Comparison of pooled and unpooled calibrations.





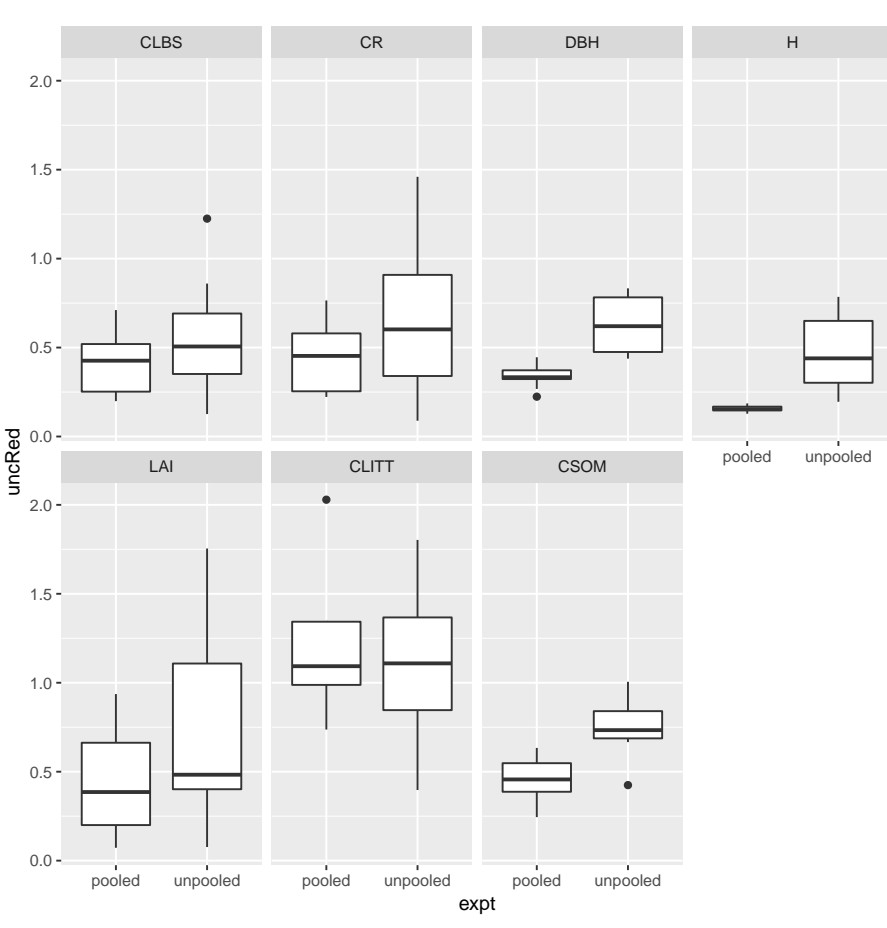

**Figure 15.** Ratio of posterior to prior 5 to 95 quantile range. Comparison of pooled and unpooled calibrations.



**Table 1.** Table of forest sites that were included in the calibration

| Site Name | Country | FluxNET Abbreviation | Type |
|---|---|---|---|
| El Saler | Spain | ES-ES1 | Pine |
| Hyytiälä | Finland | FI-Hyy | Pine |
| Sodankyla | Finland | FI-Sod | Pine |
| LeBray | France | FR-LBr | Pine |
| San Rossore | Italy | IT-SRo | Pine |
| Loobos | Netherlands | NL-Loo | Pine |
| Speulderbos | Netherlands | NL-Spe | Pine |
| Norunda | Sweden | SE-Nor | Pine |
| Skyttorp | Sweden | SE-Sk2 | Pine |
| Bily Kriz | Czech Republic | CZ-BK1 | Spruce |
| Höglwald | Germany | DE-Hoe | Spruce |
| Tharandt | Germany | DE-Tha | Spruce |
| Wetzstein | Germany | DE-Wet | Spruce |
| Renon | Italy | IT-Ren | Spruce |
| Fyodorovskoye | Russia | RU-Fyo | Spruce |
| Griffin | UK | UK-Gri | Spruce |
| Hainich | Germany | DE-Hai | Deciduous |
| Sorø | Denmark | DK-Sor | Deciduous |
| Fontainebleau-Barbeau | France | FR-Fon | Deciduous |
| Hesse | France | FR-Hes | Deciduous |
| Collelongo | Italy | IT-Col | Deciduous |
| Fougères | France | FR-Fgs | Deciduous |




**Table 2.** Selected references used for the sites included in this calibration.

| Site Name | Authors | Journal | Vol. | Pages |
|-----------|---------|---------|------|-------|
| DE-Hai | Knohl et al. (2003) | Agric. Forest Meteorol. | 118 | 151-167 |
| DK-Sor | Pilegaard et al. (2003) | Boreal Environ. Res. | 8 | 315-333 |
| FR-Fon | Davi et al. (2006) | Agric. Forest Meteorol. | 139 | 269-287 |
| FR-Hes | Granier et al. (2008) | Ann. For. Sci. | 64 | n° 704 |
| IT-Col | Scartazza et al. (2004) | Oecologia | 140 | 340-351 |
| CZ-BK1 | Sedlák et al. (2010) | Agric. Forest Meteorol. | 150 | 736-744 |
| DE-Hoe | Kreutzer et al. (2009) | Plant Biol. | 11 | 643-649 |
| DE-Tha | Grünwald and Bernhofer (2007) | Tellus | 59B | 387-396 |
| DE-Wet | Anthoni et al. (2004) | Glob. Change Biol. | 10 | 2005-2019 |
| IT-Ren | Marcolla et al. (2005) | Agric. Forest Meteorol. | 130 | 193-206 |
| RU-Fyo | Ramonet et al. (2002) | Tellus | 54B | 713-734 |
| UK-Gri | Clement et al. (2003) | Scottish Forestry | 57 | 43013 |
| ES-ES1 | Sanz et al. (2002) | Environm. Pollution | 118 | 259-272 |
| FI-Hyy | Vesala et al. (2005) | Glob. Biogeochem. Cycles | 19 | n° GB2001 |
| FI-Sod | Thum et al. (2008) | Biogeosciences | 5 | 1625-1639 |
| FR-LBr | Rivalland et al. (2005) | Ann. Geophysicae | 23 | 291-304 |
| IT-SRo | Chiesi et al. (2005) | Agric. Forest Meteorol. | 135 | 22-34 |
| NL-Loo | Dolman et al. (2002) | Agric. Forest Meteorol. | 111 | 157-170 |
| NL-Spe | Erisman et al. (1999) | Water, Air, and Soil Pollution | 109 | 237-262 |
| SE-Nor | Grelle et al. (1999) | Agric. Forest Meteorol. | 98-99 | 563-578 |
| SE-Sk2 | Lindroth et al. (2008) | Tellus | 60B | 129-142 |





**Table 3.** Average number of observations per variable. GPP model derived data was not included in the calibrations.

| Variable | Average number of obs. per site |
|----------|--------------------------------:|
| CLBS     | 1.89  |
| CR       | 1.22  |
| DBH      | 2.56  |
| H        | 2.89  |
| LAI      | 9.67  |
| CLITT    | 0.44  |
| CSOM     | 1.22  |
| NEE      | 46.56 |
| **GPP**  | 46.56 |
| ET       | 49.67 |
| WC       | 31.00 |





**Table 4.** Soil water retention and rooting depth data used to inform prior ranges of site specific parameters.

| Site Name | Wilting Point (WP) ($m^3$ $m^{-3}$) | Field Capacity (FC) ($m^3$ $m^{-3}$) | meaurement available? | Saturation ($m^3$ $m^{-3}$) | Rooting Depth (m) | meaurement available? |
|---|---|---|---|---|---|---|
| CZ-BK1 | 0.085 | 0.195 | n | 0.367 | 0.55 | y |
| DE-Hai | 0.155 | 0.243 | y | 0.589 | 0.48-0.71 | y |
| DE-Hoe | 0.175 | 0.265 | n | 0.366 | 1 | n |
| DE-Tha | 0.07 | 0.16 | y | 0.435 | 1.15 | y |
| DE-Wet | 0.175 | 0.265 | n | 0.474 | 0.44 | y |
| DK-Sor | 0.085 | 0.195 | n | 0.457 | 0.70 - 0.85 | y |
| ES-ES1 | 0.085 | 0.195 | n | 0.367 | 2 | y |
| FI-Hyy | 0.08 | 0.5475 | y | 0.573 | 0.61 | y |
| FI-Sod | 0.105 | 0.255 | n | 0.467 | 1.5 | y |
| FR-Fon | 0.085 | 0.195 | n | 0.298 | 0.8 | y |
| FR-Fgs | 0.15 | 0.405 | n | 0.563 | 0.4-1.5 | y |
| FR-Hes | 0.1 | 0.4 | y | 0.600 | 1.45 | y |
| FR-LBr | 0.03 | 0.17 | y | 0.445 | 1.2 | y |
| IT-Col | 0.185 | 0.29 | y | 0.603 | 0.3-1.5 | y |
| IT-Ren | 0.135 | 0.335 | n | 0.613 | 0.5 | y |
| IT-SRo | 0.0475 | 0.0925 | y | 0.382 | 0.5-3 | y |
| NL-Loo | 0.025 | 0.124 | y | 0.251 | 1 | n |
| NL-Spe | 0.085 | 0.195 | n | 0.367 | 1 | y |
| RU-Fyo | NA | NA | n | NA | 0.3 | y |
| SE-Nor | 0.05 | 0.17 | y | 0.395 | 0.7 | y |
| SE-Sk2 | 0.055 | 0.175 | n | 0.373 | 1 | n |
| UK-Gri | 0.335 | 0.435 | n | 0.620 | 1 | n |

**Table 5.** Table of coefficients of variance and minimum $\sigma$ used to estimate uncertainty in the likelihood function.

| | NEE | ET | WC | H | DBH | LAI | CLBS | CR | CLITT | CSOM | Rh | $NO_2$ | NO |
|---|---|---|---|---|---|---|---|---|---|---|---|---|---|
| Cv | 0.3 | 0.3 | 0.3 | 0.2 | 0.2 | 0.3 | 0.2 | 0.5 | 0.5 | 0.3 | 0.5 | 0.5 | 0.5 |
| min $\sigma$ | 0.5 | 0.3 | 0.07 | | | | | | | | | | |