# Peer review of "Calibrating a process-based forest model with a rich observational dataset at 22 European forest sites."

_Biogeosciences, 2018_

## Referee Comment (RC1) · Anonymous Referee #1 · 16 May 2018

The model presents the calibration of a forest model, BASFOR, using a Bayesian MCMC algorithm, to data from a variety of sources at a range of European forest sites. The authors seek to identify which datasets improve model fit as well as finding the best way to use data spread across multiple sites. This kind of study is generally very useful in pinpointing the critical data needs for model calibration as well as finding model knowledge gaps.

However, I find that the paper is very poorly written. There is very little detail on the model structure and the way that the data was collected and processed. The results section lacks detail and actual values for model fit providing only vague statements

about the results. The discussion and conclusion are largely a repetition of the results without going into any depth. In any case, any discussion would be hard to follow without knowing more about the model and the parameters which were fitted, a fact which I am afraid is reflected in my detailed comments, which mostly focus on the first part of the paper.

I find the introduction and discussion have a very narrow focus, referring only to process-based forest models, and ignoring the very very large range of studies that calibrate terrestrial biosphere models or land surface models and ask similar or identical questions to the current study.

A few topics that I would like to see discussed more in depth are: Which parameters are constrained by the different datasets used? Are the parameters correlated? Is there equifinality? Are the inconsistencies between the data and models due to the model structure or the data used? Or the fitting algorithm? Several times in the discussion it is mentioned that there are inconsistencies between the different datasets? Why is this? Can this be improved? What is the path forward for future model calibration? What are the data needs?

Detailed comments

P2 l15 I find the discussion of the previous use of Bayesian methods for data assimilation very narrow. Whilst these references might cover the previous studies that constrain forest models there is a much wider pool of studies using terrestrial biosphere model and mand surface models which use these techniques.

P4 L15 Was the soil data measured for this study? If so, more details of the methods are needed. If not, please provide a reference.

P4 l16 What is the annual integrated metric you have used? Is it a mean or a sum or something else?

P4 l18 "Data from CEIP and other project (e.g. FLUXNET) databases as well as publications " More details are needed here: which other databases? What data was taken from where? This phrasing is repeated in subsequent sections too, my comment applies to all

P5 l1 This model description is inappropriately short. At least some basic equations or a model schematic are needed for the reader to understand how the model and data go together.

P5 l16 What do you mean by replicated? Did you use a mean seasonal cycle from the available data? The common practice is to use climate reanalysis data for time periods when local met data is not available. You at least need to explain why this was not done.

P5 l18 Due to the poor description of the model it is difficult for me to understand why this data is needed.

Sections 2.3.3-2.3.5 These all refer to driver data and should be put into that section, they d not need their own subsection. I do not understand the need for the detailed description of N deposition, much more detailed than for any other data. Does your model have a particular focus on N deposition?

P7 l1 "we chose to calibrate nearly all of the parameters" Which parameters exactly did you calibrate?

P7 l10 The forest types chosen, pine spruce and deciduous, need more explanation. Generally, models would choose to use standard plant functional types. I understand that with the focus on local modelling you might want to use species as categories but then why use 'deciduous'?

P8 l7 How did you aggregate this data?

Section 3 Phrasings like 'closer' and 'further away' need to be more quantitative, include for example use the RMSE

Section 8 Conclusions are generally text and not a bullet point list

Figures 1 and 2 Please add y-axis labels with variable names and units. Also, it would be great to see both fluxes and stocks at both sites. Figures 5-8 Figures should be understandable from the caption alone without reference to other section. Also, another missing Y label

Tables 1 and 2 It is common practice to report site name, geographical coordinates and some other site information such as forest age and climate (e.g. mean annual temperature, mean annual precipitation), maybe soil type. References need to take only one column since I assume everything is included in the bibliography and we really do not need all this information in a table

---

## Author Comment (AC1) · 24 May 2018

We thank the reviewer for their efforts. This is our initial response to the more substantive points raised – a more detailed point-by-point response will follow later. The reviewer asked for more information about the model BASFOR, the data used, and the calibration methodology including details of the parameters.

As for model detail, we think we struck the correct balance with a concise description in the text, since the focus of the paper is not the model itself but the consequences of using rich datasets in calibration. The model is not new, it has been published (three papers are listed), and we provide full information online

(https://github.com/MarcelVanOijen/BASFOR as quoted in the text) where the model itself as well as a 33-page user guide can be downloaded.

The input and calibration data that we used were not measured specifically for our study. They were measured at individual forest sites, mostly as part of larger thematic international projects such as CarboEurope and NitroEurope. Detailed descriptions of the measurements are thus available in other publications, and we provided full bibliographic information for each site in Table 2. It would lengthen our text unnecessarily to copy measurement information from these sources into our paper.

Details of the parameters calibrated in this study are given in tables in the supplementary material, as we mentioned in sections 2.4.2 and 2.4.3. However, as stated in the Introduction, our focus in this paper was on the impact of Bayesian calibration (BC) on model predictive capacity and "in particular, which observational datasets were most effective in reducing uncertainty in model predictions and data-model differences." We also explained in the Introduction why we did not clutter the paper with an analysis of parameter distributions: "Since model parameterisations are specific to models we will present results showing how model output uncertainty and model-data differences changed as a result of calibration rather than how the model parameter uncertainty changed as a result of the BC."

In contradiction to the reviewer's comments, we presented our results in considerable detail. Using ratios of RMSE and quantile ranges, we quantified the influence of the calibration on model-data differences and on uncertainty. We provide full quantitative information in the figures, while the text gives a more qualitative description in keeping with the focus of the paper.

We also stress that the final sections of the paper are no mere repetition of the Results, the opposite is the case. We used the standard approach of presenting results without interpretation in their own section, leaving the analysis to the Discussion and the Conclusions. In these latter sections, we analyse the key issue: How effective are

the different observational datasets for reducing model-data differences and uncertainties? We conclude that: "Sparse plant and soil stock observations were more important for reducing model-data differences and uncertainty in above and belowground carbon pools than more plentiful carbon and water flux data" and we point out that there are exceptions for "ecosystem variables where only very few uncertain observations were available." Our second main question is addressed there too: "Are separate calibrations at forest sites more effective for improving model fit to data and reducing uncertainty than multi-site calibration?" Our conclusion on that question is that: "While separate calibrations at each forest site generally reduced model-data differences more than calibrating at all the sites together, parts of the ecosystem that were very sparsely observed benefited more from multi-site calibration." We believe these findings were properly discussed and are important.

---

## Referee Comment (RC2) · Anonymous Referee #2 · 31 May 2018

In general, this paper is a bit long and not focused enough as to what are the main hypotheses that you want to test, what you found and why it all matters. (There's a lot of text devoted to what you did, and what happened, but it's not crafted into a coherent story.) Even the question of pooled vs. un-pooled calibration across sites has some mixed results, and it's also not clear that the "richer" dataset used here as compared to previous studies helped you to answer questions that couldn't be answered previously.

Other major and minor comments are listed below, followed by comments on tables & figures.

Major comments

The abstract is currently too long and meandering and should be shortened and focused. Some terminology is also unclear until after reading the paper, e.g. "single dataset calibrations". It's not clear what datasets you're referring to. The abstract also suggests the use of Bayesian hierarchical calibration to realize the benefits of both the pooled and un-pooled approaches. Since this method is not specifically applied in this study, the authors should clarify that this method may be useful in future work. (Is there any chance to include a small case study with Bayesian hierarchical calibration, perhaps in SM? Or just do a test and describe in a few sentences the results in the main text?)

It's not clear if the same observational data is first used for calibration, and then for validation (model-data differences). Wouldn't this be a non-independent use of the data for validation/ evaluation of the model? It appears to be so given your statement on page 16, line 6 that "Most obviously, reducing model-data differences is most effective when the data used in the comparison are also included in the calibration". The authors need to better justify this double use of the data for both calibration and validation.

The term "rich dataset" is somewhat over-used since this is a vague term. Should specify quantitatively how many more ecosystem variables and sites you are using than was used in prior studies. Also, consider making the paper title more specific in terms of the dataset used for calibration. What is it that you can do with this dataset and the BASFOR model that wasn't possible in previous studies?

Throughout the results sections (4 to 6) , there are a large number of statements that describe modeling results without offering any explanation as to why the phenomenon occurred. Please keep in mind that the reader knows less about this topic than you do, even if you discuss the results more generally in the Discussion section. You don't need to describe every aspect of your results, only the important ones that help to tell a story. If there is no significance in a particular result, consider removing the statement. Some examples below:

[Figure]

* p. 11, lines 13 to 19, entire paragraph starting with: "For the low frequency variables, model outputs at sites NL-Spe and IT-Ren generally moved furthest away from the observations, relative to the prior mode..."

* p. 12, line 1: "There is also a much stronger relationship between the number of observations included in the calibration and the reduction in uncertainty from prior to posterior than was true for RMS deviation changes after calibration. It is also noted that uncertainty reduction was less where the variance in time of the model output increased significantly from prior to posterior."

* p. 12, lines 21 to 24: paragraph starting with "Not all the calibrations improved the comparison against observations. Indeed, the "ET" calibration increases model differences...."

* p. 13, lines 8 to 10: "The "ET" calibration is most likely to increase differences with observations after calibration..." Physiologically, why would an ET-only calibration reduce above-ground C sequestration?

The authors should describe the rationale for looking at the two different metrics of model quality, 1) the normalized RMS deviation (ratio between posterior and prior) and 2) the range of the 95th to 5th quantiles (ratio between posterior and prior). What different sorts of information is each metric providing? In terms of rationale, it appears that 1) gets at model fit to data, whereas 2) is more of the mathematical representation of uncertainty. What does it mean when model uncertainty goes down, but model fit to data gets worse? The authors should discuss this, e.g. in last paragraph of page 15, although this is later touched upon in section 7.1.2. Also, the word "uncertainty" is used loosely throughout the paper but it should really be clarified early on in the methods and then used in a precise way going forward.

In general, there is a lot of detail in the Methods section which could be moved to the supplemental material to help improve the flow of the paper. For example, most of section 2.4, especially sections 2.4.5 and 2.4.6 on the optimization procedure, are

details that detract from the main story of the paper. Also, in the Methods section, it makes more sense to first present the model (i.e. BASFOR) and then describe the observational data used to calibrate and validate it, i.e. move sections 2.1.2 and 2.1.3 after 2.3 (model driving/ input data).

What is the rationale for looking at the 6 different datasets for calibration? If you have all this data, wouldn't you typically use all of it (both high and low-frequency)? Are you trying to make recommendations for cases when other researchers have less data available to them? The authors should try to better explain the rationale for these tests, rather than just trying out a bunch of stuff and then reporting the results.

When discussing the relative benefits of pooled vs. un-pooled calibrations, please keep in mind that it matters where you are planning to apply the model. If the model is mainly used at the sites included in the calibration, un-pooled parameters may make more sense. Also, if the areas you are extrapolating to can be well-represented by one or a few sites included in the calibration, un-pooled parameters optimized for just those sites may be best. If areas where the model is applied cannot be neatly classified into types as represented by the calibration sites, then a pooled model that captures more variability across ecosystems may be better. In this study, it looks like you don't apply the model to sites not included in the calibration; so therefore, this consideration is not discussed, but likely relevant for "real-world" applications of the model.

If this study had access to a richer dataset with more variables describing different parts of the ecosystem than previous studies using BASFOR, how do the MAP parameters from the "all data" calibration compare to those from previous studies? Can you learn something about ecosystems and/ or model performance from your best model?

Minor comments

Should try to simplify & clarify terminology throughout the paper when comparing prior vs. posterior (i.e. MAP) and come up with shorter names for the various metrics you use to assess model quality. It's not easy to read or follow now. For example, in the

caption to Figure 4: "Ratio of the posterior to the prior of the range of the 5th to the 95th quantile". Maybe simplify to something like: "uncertainty reduction from posterior to prior", and then refer to the section where this is explained in detail. It also would be useful to provide 2 simple equations in the methods section (2.6) to clarify what each of the two uncertainty metrics (normalized RMS deviation and the range of the 5th to the 95th quantile) are showing.

There are lots of long sentences without commas throughout the paper, which makes it hard to understand the flow of the sentences and arguments. For example, p. 11, line 27: "Of these model outputs for NL-Loo was one of the closest sites to observations prior to BC." Also, page 12, lines 18-20: "Model derived GPP was not included in the calibration so in this case the largest decreases are found when all the calibration data are included although as might be expected inclusion of NEE is the next most important dataset for decreasing model and data differences". Many more examples throughout the paper.

p. 5, line 16: "the (weather) data were replicated backwards in time based on the available time series. . ." Can you please explain how you replicate weather data backwards in time, and the potential modeling errors associated with this replication?

p. 5, lines 24-25: "A figure showing the stand history reconstructions used is given in the supplementary material", I don't see this in the SM. p. 6, lines 24-25: "We chose the Beta distribution." Why? Should try to offer brief justification for modeling choices.

Why did the authors choose to use site-specific priors for the water retention curve, rooting depth and initial soil and litter carbon values (p. 7, lines 11-12)? Was it availability of data, or knowledge of ecosystem variability?

p. 8, lines 6-7: Why is a 30-day average of model output more reliable than daily? Why not 3-monthly or annual? 30 days seems kind of arbitrary.

p. 8, line 8: Remind us what sigma is again. Is this measurement error?

p. 8, line 24: "In this study, we made three kinds of calibration". Later, you refer to the multi-site tests with 6 different calibration datasets. Can you clarify here how the three kinds are related to the 6 datasets? Will help reader to follow along later.

Does the pooled calibration include pine, spruce and deciduous sites? (It seems like it based on page 9, line 14: "... identical to the default calibration labelled 'All' above"). Why not restrict this pooled test to just the pine sites? Why did you choose pine for this pooled vs. un-pooled set of tests?

Page 9, line 27: Please explain what "burn-in" is.

Sections 3.1 and 3.2 (page 10), isn't it pretty obvious that the calibration would bring the posterior closer to the observations than the prior? How representative are the two sites that you chose in terms of the calibration performance of other sites?

p. 16, line 24: "In general, calibration data was less effective in reducing model-data differences when there were larger inconsistencies present." This is a pretty vague sentence, inconsistencies between what and what?

p. 17, line 18: "Further, large reductions in uncertainty also occurred when the observations included in the calibration had a tendency to increase model-data differences." Shouldn't this be to decrease model-data differences? Shouldn't model uncertainty theoretically go down when model/ data fit improves?

p. 19, lines 12-13: "suggesting a strong relationship between the underlying processes as represented in the model." This seems like a self-evident statement that the ecosystem variables included in the model are likely correlated with one another. Can you be more specific here?

Tables and Figures

It would be good to reduce the number of figures in the paper, e.g. by combining high and low frequency variables into a single plot (i.e. Figs 5/6, 7/8, 9/10, 11/12), and then indicating high vs. low frequency using background shading or labels. The authors

may also consider just showing one metric or another rather than both in the main text (moving the alternative metric to SM). Given a "storyline" in the paper, how does each figure help to illustrate an aspect of this story? The figure captions should also indicate what are "good" values for the metrics being shown, or otherwise, how to interpret them, i.e. that values lower than 1 imply some amount of uncertainty reduction.

Figures 1 and 2: how did you choose which sites to show? Shouldn't the posterior be closer to the observations by definition? Maybe combine Figures 1 and 2, and/or move to SM? These figures could also use a legend. Should also specify which calibration is being shown. Is this "all data"?

Figures 3, 4 and 13: It would be helpful to show means for each row and column in these plots, at least across sites for each variable. Figures 3 and 13 are also small and hard to read. For Figure 3, consider flipping each sub-figure and then stacking the two up and down? Might make more space for bigger labels. Again, which calibration are you looking at?

Figures 9 and 10: where is the spread in the boxplots coming from? Is this the spread across sites? Should specify this in caption, and also that you're looking specifically at pine sites here.

Table 3: this is mentioned in text after Table 4, therefore the tables should be re-ordered. Also, please include the full name for each variable, which will help reader to follow along in the text. It would also be nice to show the timescale and the start and end years for each variable. Why is GPP in bold?

Table 4: Do the "measurement available" columns refer to the previous columns? Should clarify.

Table 5: What is the sigma value referred to here? Should define. Also, why are Rh, NO2 and NO included here, but not in any of the results?

Supplemental material

Why is Figure 2 in between Tables 1 and 2?

Figures 3, 5 and 6: what do you mean by "ancillary observations" in the figure captions? How are these different from ordinary "observations", e.g. in Figure 4 caption?

---

## Referee Comment (RC3) · Anonymous Referee #3 · 11 Jun 2018

General comments:

The authors calibrate the BASFOR forest model using various data and analyze what data best constrains the posterior predictive model uncertainty. The type of work presented is important even though new methodology is not presented. The novelty of the paper comes from using more data and in a more varied setting than was done before.

I will comment on the calibration part in these comments, since I'm not an expert of forest modeling.

Whereas the work does have merits, it is seriously lacking in detail, and many design choices seem rather arbitrary. There are no formulas describing what was done - even

though Bayesian model calibration is a mathematical exercise.

There are several other issues as well, the most important ones being lack of cross validation, model and model parameter descriptions are virtually missing, and although Bayes' rule is described, a probabilistic explicit observation model is not given. Details of the MCMC experiment are not properly described either.

Posterior parameter values are not described, not even in the supplements. That belongs to the main text. I would want to see full descriptionos of the parameters, priors, and posteriors in the main text, and a discussion referring to model equations about what the results of the calibration mean.

The manuscript is too long - there is a lot of repetition, and many of the conclusions are quite obvious. Such repetition should be reduced to a minimum. In addition to this, there are still very obvious editing errors, which need to be corrected. Not all of them are listed below, since there are too many.

For these reasons, a major revision is still needed.

Some more detailed comments below:

Specific comments:

1. One general issue with model calibration is, that once calibration is done, it can be used to inform where the model can be improved - where it does not perform well and where it does not. I don't understand why this aspect is not discussed. Please add this aspect throughout, especially in the discussion.

2. Please make the abstract more concise and articulated: what's new in the work and why it is important. At the moment it is a little long and unclear. And please check the language/style.

3. One enormous source of uncertainty is biases in input data. Could you please comment on this aspect.

[Figure]

4. p 4 l 30: The 1m rooting depth seems pretty arbitrary, and I'd guess it does affect the results. Why was this value chosen, and if it's not known, why isn't it included in the calibration? How much are the results affected?

5. p 5 section 2.2. You need to give the model equations, pointing out the parameters you are calibrating in those equations. This is crucial - at the moment it is impossible to say what happened in the calibration, when the parameters are not described at all. The manuscript should be self contained in that by reading it one does understand what happened.

6. p 5 l 16: "replicated backwards...?" what does this mean? Please describe in a detailed and compact way what you did, preferably with equations.

7. p 5 l 22: planting density assumption: yet again an important source of potential bias / uncertainty that is not discussed at all. Please include in a discussion of the input data uncertainty.

8. p6 l24: What parameters where used for the Beta distribution? Why? How did you use literature data to obtain the priors? Tables in supplements should be moved / summarized in the main text, with references to how the priors were chosen.

9. p6 l28: Why the covariance structures are not shown? The pairwise posterior marginals contain the most interesting information: What directions in the parameter space are constrained and what are not; are the correlations linear; are the distributions unimodal and close to Gaussian etc. An important part of the analysis should be model parameter (and hence process) identification and finding information about what data constrains what processes. The covariance structure is an important key to this analysis. Please include a figure of the 2d-marginals with probability contours (using e.g. Gaussian KDE), and discuss.

10. p7 l16: In a manuscript describing Bayesian calibration of a model, the prior values most definitely belong to the main text, not to the supplement. Please include it here.

I'd like to read the prior values for different forest types from a table. Justification of the prior distributions used should be included.

11.  p7 l26 "Prior uncertainty was set at 20% of mode value..." But isn't your "prior uncertainty" set by the parameters of the Beta distribution? Also, in any case the 20, 30 and 40% values are arbitrary, and this should be stated explicitly as a potential source of error. Please clarify.

12.  p7 l30 ..."at a given parametrisation" should rather be "at a given point in the parameter space" or something along those lines

13. p7 l31 "uncertainty about random data error", do you mean "data uncertainty". The data uncertainty should be discussed more widely, see later.

14. p. 8 l. 3. What is the observation equation (like y_t = M(x_0,t;theta) + eps, where eps $\sim$ some distribution)? How does the averaging come into play in that equation? And again: "measure of random error about ith data point". Do you mean measurement error? Or something else? Please discuss the merits of the chosen likelihood function, what does it mean? (also see next item)

15. (same) It looks to me like you are treating the residuals as independent. Is this a reasonable assumption? An alternative would be to fit a time series model (like an AR/ARMA/ARIMA etc model or such) to produce wider posteriors. You should analyze the residuals and verify that whatever probabilistic observation model description you are using, your residuals in the end conform with your error model (please show histograms of the residuals and autocorrelation functions in the supplements). If you are not able to do this, you must aknowledge that all the ranges in the figures and the scaling ("steepness") in the posterior probability distributions is arbitrary. At this point, what you have left, is then the covariances of both parameters and predicted errors / quantities, and discussing those would still be valuable. In the best case scenario, you should add both.

16. p8 l7: So you are predicting monthly values? Clarify and see previous point.

17. section 2.4.5. Details are missing. What was the proposal distribution for MCMC? Where is the picture of the chain? (please include). How long was the burn-in? What was the acceptance ratio? Effective sample size? And was there a reason to go with just Metropolis? Usually e.g. Adaptive Metropolis works a lot better (unless if you happen to a priori know what the optimal proposal is)

18. p8 l. 24. The lonely sentence could be left out. Throughout the text there are lots of these types of sentences, and many of them could be just removed.

19. p 10 l1. I'd really like to see the corresponding contours on the pairwise 2d marginals instead / in addition.

20. Generally about results: Of course a fitted model fits data better. To know anything of how successful / good the calibration is, a k-fold cross validation (with a suitabke k) should be done instead and all the results should be reported for those. That would give information about how good the calibration is for _predictive_ purposes. At the moment the results read a little like "the model was fitted to data and after fitting, the model fits the data better". With cross validation the results would be significantly more valuable and interesting.

21. p. 10 l. 18: "Model output uncertainty is reduced from prior to posterior". This is not interesting without the cross validation - the result could be very site-specific and for predictive purposes the result could be in many places different. This being said, there is some value in the analysis, that calibrating the model using data x reduces error in variable y. But without including the model equation this is hard to see. Please, discuss these types of results with references to the model equations. Also, I would like to see what the expected value of the change of this error is.

22. p. 11 l. 2 Are you sure this is the right metric? What if you get some really bad outliers but mostly good behavior? Is this then an acceptable model? If such behavior

never occurs, the approach can be ok.

23. section 4.1 would be meaningful with the cross validation, but not so much otherwise

24. p. 12 l. 1-2: I have difficulty understanding the sentence.

25. Section 5.1. Would love to see what these results mean with some short and concise discussion. A time series of the predicted values before and after calibration would be valuable.

26. General: the observation set is advertised to be "rich" a few times too many

27. p 15 l. 20. In practice there are always inconsistencies with real data. I'd say such results tell about weighting of the different variables in the likelihood function (choice of error model).

28. p. 15 l. 21, 25: We will return... & We will discuss... could compactify the text a little here and leave these out.

29. p. 15 l. 28 "mathematical probabilities", rather say "a probabilistic model" and describe the model properly as I mentioned earlier.

30. p. 16. Lots of text for the content. Simplify and remove repetition.

31. sect. 7.1.1 generally: without the equations it is difficult to say much about whether the results are just obvious or if there is something interesting here. The discussion is not very helpful either. Please refer to model equations to explain model behaviour.

32. p. 18 l. 32. You recommend using a hierarchical model for describing the parameters, but you are not doing a hierarchical modeling yourself. Why? Also, if you were to hierarchically model the parameters, you'd need to have a way of predicting the parameters for predictive purposes. How would you do that?

33. p. 19-20. I would like to see a more compact Conclusions section, and also I'm not

sure that the bullet point style is a good idea.

34. p. 19 l. 22. What's the role of the prior?

35. p. 19 l. 27. It is unclear to me how model structural errors were represented in your work. How did you quantify that "uncertainty is not reduced inappropriately"?

Fig. 2: The wavy shapes look strange - where did that come from?

Figures: Merge 5&6, 7&8, 9&10, 11&12, 14&15; check legend style/formatting and maybe optimize the general presentation, with improved captions. Explain in words what is seen in pictures.

Technical corrections:

grammar: e.g. p1/l12 inclusion...were => was

p4 l 29 www address to references? (check journal style, but it does not look good.)

p6 l6 on: sentences don't have verbs.

p6 l19: "probability calculus" => Bayes' rule

p8 l 26 "of" missing most likely

p13 l30 2nd sentence, sounds a little strange.

---

## Author Comment (AC2) · 29 Jun 2018

**Authors response**

Firstly, we would like to thank the referees for their efforts and for their ideas on how the paper can be improved. We were pleased to see that they regard the work as "very useful" and "important". But we also agree that the paper is too long which makes it harder to identify the main story and key aims.

The two key elements of the paper are: (1) the availability of a uniquely rich dataset of European forests, and (2) what that dataset makes possible: identification of the most

informative data for calibration of a process-based forest model. We analyse how including or excluding different data types affects model predictions after calibration. This information may help design future measurement campaigns and model calibrations. The referees missed these key aims and instead thought that the paper was about the uncertainty quantification and analysis of the model BASFOR. In our revision, we shall shorten the paper to sharpen and improve the story; making the key aims more apparent. We will also explicitly state – in the Introduction – what the paper is not about, i.e. uncertainty quantification and analysis of BASFOR.

Response to Referee 1

The following comments address similar issues so we answer them together.

"*The model presents the calibration of a forest model, BASFOR, using a Bayesian MCMC algorithm, to data from a variety of sources at a range of European forest sites. The authors seek to identify which datasets improve model fit as well as finding the best way to use data spread across multiple sites. This kind of study is generally very useful in pinpointing the critical data needs for model calibration as well as finding model knowledge gaps.*"

"*However, I find that the paper is very poorly written. There is very little detail on the model structure and the way that the data was collected and processed. The results section lacks detail and actual values for model fit providing only vague statements about the results. The discussion and conclusion are largely a repetition of the results without going into any depth. In any case, any discussion would be hard to follow without knowing more about the model and the parameters which were fitted, a fact which I am afraid is reflected in my detailed comments, which mostly focus on the first part of the paper.*"

"*P5 l1 This model description is inappropriately short. At least some basic equations*

*or a model schematic are needed for the reader to understand how the model and data go together.*"

"*P5 l18 Due to the poor description of the model it is difficult for me to understand why this data is needed.*"

- We answered these comments in our previous response (AC1).

The following comments address similar issues so we answer them together.

"*I find the introduction and discussion have a very narrow focus, referring only to process-based forest models, and ignoring the very very large range of studies that calibrate terrestrial biosphere models or land surface models and ask similar or identical questions to the current study.*"

"*P2 l15 I find the discussion of the previous use of Bayesian methods for data assimilation very narrow. Whilst these references might cover the previous studies that constrain forest models there is a much wider pool of studies using terrestrial biosphere model and mand surface models which use these techniques.*"

- In this paper we have chosen to focus on forest ecosystem models rather than any other models such as land surface or global vegetation models. We will however add a couple of references from other modelling work to the revised paper.

"*A few topics that I would like to see discussed more in depth are: Which parameters are constrained by the different datasets used? Are the parameters correlated? Is there equifinality?*"

- As stated in our previous response (AC1) our focus in this paper is not to answer these kinds of questions interesting though they may be in their own right.

"*Are the inconsistencies between the data and models due to the model structure or the data used? Or the fitting algorithm? Several times in the discussion it is mentioned that there are inconsistencies between the different datasets? Why is this? Can this be improved?*"

- The text to which the reviewer is referring is "Secondly, the single dataset calibrations highlighted where there were inconsistencies either between the different observational datasets or between the observations and the model or both (subsequently referred to in the text as just "inconsistencies")."
- The contribution of model structural error and data systematic biases to these "inconsistencies" is very interesting and informative but is not straightforward to disentangle and should be a focus of future work.
- It is highly unlikely that the MCMC algorithm itself is contributing to the inconsistencies that were found.

"*What is the path forward for future model calibration? What are the data needs?*"

- The path forward is stated in the final conclusions of section 8.1 and 8.2. "Uncertainty about model structural errors and systematic observational biases should be represented in Bayesian calibration so that uncertainty is not reduced inappropriately by calibration." and "The best compromise is to allow variation in model parameters between different sites but to share information across sites to improve models. Such a compromise is possible with Bayesian hierarchical calibration."
- In terms of data needs we conclude that "Sparse plant and soil stock observations were more important for reducing model-data differences and uncertainty in above and belowground carbon pools than more plentiful carbon and water flux

data. Except for ecosystem variables where only very few uncertain observations were available." This would implicitly suggest that this data while sparse is important. A more explicit statement about the need to collect stock observations in the future will be added in the revised paper.

"*P4 L15 Was the soil data measured for this study? If so, more details of the methods are needed. If not, please provide a reference.*"

- Soil fluxes were measured as part of various research projects. References for each site are given in Table 2. We will add reference to this table in the revised paper.

"*P4 l16 What is the annual integrated metric you have used? Is it a mean or a sum or something else?*"

- We agree that the current text is too vague. We will change the text to make it clear that we employed annual averages.

"*P5 l16 What do you mean by replicated? Did you use a mean seasonal cycle from the available data? The common practice is to use climate reanalysis data for time periods when local met data is not available. You at least need to explain why this was not done.*"

- By replicated we mean that the whole available (5-20 yr depending on the site) observed weather dataset is copied backwards in time as many times as is needed to fill the X-year time-series to cover the whole lifetime of the forest. We think that using this local site based data is preferable to a modelled time series from climate re-analysis which would introduce significant downscaling errors. By

doing this we are aware that we ignore climate change which may be significant for the older forests.

"*Sections 2.3.3-2.3.5 These all refer to driver data and should be put into that section, they d not need their own subsection. I do not understand the need for the detailed description of N deposition, much more detailed than for any other data. Does your model have a particular focus on N deposition?*"

- Breaking up the "Model driving/input data" section into subsection increases the clarity for the reader over a single large section.
- The subsection for N deposition was longer since the creation of this input was more complex than for the other inputs.

"*P7 l1 "we chose to calibrate nearly all of the parameters" Which parameters exactly did you calibrate?*"

- In general (pine and spruce) we grouped the forest sites based on genus. For the deciduous sites there were too few sites in each genus so all the deciduous sites were grouped together.

"*P8 l7 How did you aggregate this data?*"

- As we mentioned in the text, we "aggregated all the daily data (carbon and water fluxes and soil water content) to a 30 day average."

"*Section 3 Phrasings like 'closer' and 'further away' need to be more quantitative, include for example use the RMSE*"

- The plots give the full quantitative information. The text provides a descriptive summary of what the plots show.

"*Section 8 Conclusions are generally text and not a bullet point list*"

- Bullet points give a clearer summary of the main conclusions of the paper than a paragraph of text so we disagree with this comment.

"*Figures 1 and 2 Please add y-axis labels with variable names and units. Also, it would be great to see both fluxes and stocks at both sites.*"

- These figures will be removed in the revised version of the paper.

"*Figures 5-8 Figures should be understandable from the caption alone without reference to other section. Also, another missing Y label*"

- We will expand the figure captions in the revised version as suggested.
- The y-axis quantity is the ratio as stated in the caption.

"*Tables 1 and 2 It is common practice to report site name, geographical coordinates and some other site information such as forest age and climate (e.g. mean annual temperature, mean annual precipitation), maybe soil type. References need to take only one column since I assume everything is included in the bibliography and we really do not need all this information in a table*"

- More information on site characteristics will be added in the revised version.
- The reference information will be provided in a single column in the revised version.

Response to Referee 2

"*In general, this paper is a bit long and not focused enough as to what are the main hypotheses that you want to test, what you found and why it all matters. (There's a lot of text devoted to what you did, and what happened, but it's not crafted into a coherent story.) Even the question of pooled vs. unpooled calibration across sites has some mixed results, and it's also not clear that the "richer" dataset used here as compared to previous studies helped you to answer questions that couldn't be answered previously. Other major and minor comments are listed below, followed by comments on tables & figures.*"

- We do state our main questions in the introduction p3 l22-29 and discuss them in sections 7.1 and 7.2 however, we agree that the paper can be shortened which will improve the focus and coherence of the main story. We will remove section 3 completely and remove material in sections 4-6 that is not returned to sections 7 and 8.
- The use of a rich dataset enabled us to state in the abstract that "Our results suggest that use of calibration data representing just a few aspects of the ecosystem could be problematic, since improved model-data fits for the parts of the system represented by the data could be at the expense of other part of the system, where the model-data fit worsened." and in the conclusions "Calibrations including data that represents a greater variety of ecosystem variables, is a more stringent test of an ecosystem PBM, since the model has to stay consistent with a greater diversity of the observed ecosystem, for there to be a good fit to the observations after calibration." However, we will make a more explicit link between the "rich" datasets used in this study and these messages.

"*The abstract is currently too long and meandering and should be shortened and focused. Some terminology is also unclear until after reading the paper, e.g. "single*

*dataset calibrations". It's not clear what datasets you're referring to. The abstract also suggests the use of Bayesian hierarchical calibration to realize the benefits of both the pooled and unpooled approaches. Since this method is not specifically applied in this study, the authors should clarify that this method may be useful in future work. (Is there any chance to include a small case study with Bayesian hierarchical calibration, perhaps in SM? Or just do a test and describe in a few sentences the results in the main text?)*"

- We agree that the abstract can be shortened while retaining the main points that we wish to make.
- We will replace "single dataset calibrations" with calibrations that included a single one type of data in the text.
- We will change the wording to "These results support the investigation of Bayesian hierarchical calibration in future work which allows..."
- Applying hierarchical methods to complex process-based models is currently not straightforward. This is an active area of research for us.

"*It's not clear if the same observational data is first used for calibration, and then for validation (model-data differences). Wouldn't this be a non-independent use of the data for validation/ evaluation of the model? It appears to be so given your statement on page 16, line 6 that "Most obviously, reducing model-data differences is most effective when the data used in the comparison are also included in the calibration". The authors need to better justify this double use of the data for both calibration and validation.*"

- We agree that it would be problematic if we were to try and use the same data for calibration and for model evaluation/validation. However, our aim here is not

to evaluate/validate the model but instead to analyse "the influence of the calibration on the model predictions and in particular, which observational datasets were most effective in reducing uncertainty in model predictions and model-data differences." We are focused here on the informativeness of different kinds or data. This is already stated in the introduction but we will add text to make it clear that our purpose was not model evaluation and validation. Indeed we discuss and recognise in the paper that model structural errors and systematic data errors influence our results so future diagnostic evaluation and inclusion of model and data errors in BC will improve on the results presented here.

"*The term "rich dataset" is somewhat over-used since this is a vague term. Should specify quantitatively how many more ecosystem variables and sites you are using than was used in prior studies. Also, consider making the paper title more specific in terms of the dataset used for calibration. What is it that you can do with this dataset and the BASFOR model that wasn't possible in previous studies?*"

- While there is already information in the introduction p3 on how our calibration dataset compares with those used in previous studies we agree that this point could be sharpened through the inclusion of a comparison table in the revised version.
- We answer the comment "what we were able to say with a richer dataset" above but agree that this point needs to be made explicitly.

"*Throughout the results sections (4 to 6) , there are a large number of statements that describe modeling results without offering any explanation as to why the phenomenon occurred. Please keep in mind that the reader knows less about this topic than you do, even if you discuss the results more generally in the Discussion section. You don't need to describe every aspect of your results, only the important ones that help to tell a*

*story. If there is no significance in a particular result, consider removing the statement. Some examples below:*"

"*p. 11, lines 13 to 19, entire paragraph starting with: "For the low frequency variables, model outputs at sites NL-Spe and IT-Ren generally moved furthest away from the observations, relative to the prior mode. . ."*"

"*p. 12, line 1: "There is also a much stronger relationship between the number of observations included in the calibration and the reduction in uncertainty from prior to posterior than was true for RMS deviation changes after calibration. It is also noted that uncertainty reduction was less where the variance in time of the model output increased significantly from prior to posterior."*"

"*p. 12, lines 21 to 24: paragraph starting with "Not all the calibrations improved the comparison against observations. Indeed, the "ET" calibration increases model differences. . .."*"

"*p. 13, lines 8 to 10: "The "ET" calibration is most likely to increase differences with observations after calibration. . ." Physiologically, why would an ET-only calibration reduce above-ground C sequestration*"

- This point on improving the readability of the paper is answered above.
- We deliberately leave discussion of explanations of phenomenon to the discussion section as this is the appropriate place for this.

"*The authors should describe the rationale for looking at the two different metrics of model quality, 1) the normalized RMS deviation (ratio between posterior and prior) and 2) the range of the 95th to 5th quantiles (ratio between posterior and prior). What different sorts of information is each metric providing? In terms of rationale, it appears that 1) gets at model fit to data, whereas 2) is more of the mathematical representation of uncertainty. What does it mean when model uncertainty goes down, but model fit to*

*data gets worse? The authors should discuss this, e.g. in last paragraph of page 15, although this is later touched upon in section 7.1.2. Also, the word "uncertainty" is used loosely throughout the paper but it should really be clarified early on in the methods and then used in a precise way going forward.*"

- We agree that more can be said to clarify our rationale for the two different metrics used. For example our aim with the RMS measure is not model fit per se but rather how model-data difference changes as a result of calibrations with different datasets. The change 95th - 5th quantiles is to quantify how uncertainty in the model outputs changes after calibration with different datasets.
- We think that section 7.1.2 addresses model uncertainty going down but model fit to data getting sufficiently.
- We disagree that the term uncertainty is used loosely in the paper. We defined the change in uncertainty precisely in section 2.6.

"*In general, there is a lot of detail in the Methods section which could be moved to the supplemental material to help improve the flow of the paper. For example, most of section 2.4, especially sections 2.4.5 and 2.4.6 on the optimization procedure, are details that detract from the main story of the paper. Also, in the Methods section, it makes more sense to first present the model (i.e. BASFOR) and then describe the observational data used to calibrate and validate it, i.e. move sections 2.1.2 and 2.1.3 after 2.3 (model driving/ input data).*"

- The referees disagree about the level of detail required in the methods section. Referees 1 and 3 are asking for more detail and this referee wants material to be moved to supplementary material. On balance we think that we have the balance right with the essential details in the paper but with references provided with greater detail for readers with particular interests seeking more detail.

- We think the information in section 2.4 is needed. Sections 2.4.5 and 2.4.6 are short and contain important information.
- We think that it makes greater sense for the data section to proceed the model since the main focus of this paper is on the data rather than the model.

"*What is the rationale for looking at the 6 different datasets for calibration? If you have all this data, wouldn't you typically use all of it (both high and low-frequency)? Are you trying to make recommendations for cases when other researchers have less data available to them? The authors should try to better explain the rationale for these tests, rather than just trying out a bunch of stuff and then reporting the results.*"

- One of the main aims of this work is to quantify the informativeness of the different datasets in the calibration. The reviewer is correct that a rational for this is that by necessity we are often in a data poor situation. We are thinking about the common situation where there is plentiful eddy covariance data but only a very few carbon stock measurements are available. The six experiments are explicitly designed around this question. Is the low-frequency data, since it is more expensive to collect, valuable in the calibration? In other words, we are asking "what is lost?" where they may be very few or no calibration data for an ecosystem variable. This work also motivates the continued collection of more expensive carbon and nitrogen stock measurements. We will add these rationale to the revised paper.

"*When discussing the relative benefits of pooled vs. un-pooled calibrations, please keep in mind that it matters where you are planning to apply the model. If the model is mainly used at the sites included in the calibration, un-pooled parameters may make more sense. Also, if the areas you are extrapolating to can be well-represented by one or a few sites included in the calibration, un-pooled parameters optimized for just those*

*sites may be best. If areas where the model is applied cannot be neatly classified into types as represented by the calibration sites, then a pooled model that captures more variability across ecosystems may be better. In this study, it looks like you don't apply the model to sites not included in the calibration; so therefore, this consideration is not discussed, but likely relevant for "real-world" applications of the model.*"

- This is a useful point and is indeed not considered in this study and will be added as a consideration in the discussion.

"*If this study had access to a richer dataset with more variables describing different parts of the ecosystem than previous studies using BASFOR, how do the MAP parameters from the "all data" calibration compare to those from previous studies? Can you learn something about ecosystems and/ or model performance from your best model?*"

- As stated above this was part of the rationale for the 6 experiments with varying datasets and will be added to the revised version.
- Our focus here was on a comparison of model output variables when all the data was included versus the single dataset calibrations and we do indeed discuss what might be learnt about model structural and data systematic errors. For example the apparent conflict between the carbon (NEE) and the water cycles (ET). A MAP parameter study may reveal more/other aspects but that was not the focus of this study.

"*Should try to simplify & clarify terminology throughout the paper when comparing prior vs. posterior (i.e. MAP) and come up with shorter names for the various metrics you use to assess model quality. It's not easy to read or follow now. For example, in the caption to Figure 4: "Ratio of the posterior to the prior of the range of the 5th to the 95th quantile". Maybe simplify to something like: "uncertainty reduction from posterior*

*to prior", and then refer to the section where this is explained in detail. It also would be useful to provide 2 simple equations in the methods section (2.6) to clarify what each of the two uncertainty metrics (normalized RMS deviation and the range of the 5th to the 95th quantile) are showing.*"

- There is a balance to be struck here because in general it is good to be able to interpret plots without having to refer to the text.
- We do not think equations are needed for such simple/well known metrics such as RMS?

"*There are lots of long sentences without commas throughout the paper, which makes it hard to understand the flow of the sentences and arguments. For example, p. 11, line 27: "Of these model outputs for NL-Loo was one of the closest sites to observations prior to BC." Also, page 12, lines 18-20: "Model derived GPP was not included in the calibration so in this case the largest decreases are found when all the calibration data are included although as might be expected inclusion of NEE is the next most important dataset for decreasing model and data differences". Many more examples throughout the paper.*"

- We agree with this comment and will improve the readability of the paper by shortening sentences in the revised version.

"*p. 5, line 16: "the (weather) data were replicated backwards in time based on the available time series" Can you please explain how you replicate weather data backwards in time, and the potential modeling errors associated with this replication?*"

- We answered this question in our reply to reviewer 1.

"*p. 5, lines 24-25: "A figure showing the stand history reconstructions used is given in the supplementary material", I don't see this in the SM.*"

- This is an omission. The plot will be added to the SM in the revised version.

"*p. 6, lines 24-25: "We chose the Beta distribution." Why? Should try to offer brief justification for modeling choices.*"

- We chose the Beta distribution because as stated in the text it "is bounded and can be non-symmetric."

"*Why did the authors choose to use site-specific priors for the water retention curve, rooting depth and initial soil and litter carbon values (p. 7, lines 11-12)? Was it availability of data, or knowledge of ecosystem variability?*"

- We chose these parameters to be site-specific because soils differ between sites, whereas species (i.e. the plant parameters) are shared between sites.

"*p. 8, lines 6-7: Why is a 30-day average of model output more reliable than daily? Why not 3-monthly or annual? 30 days seems kind of arbitrary.*"

- We will add to the text that "Our choice of averaging period for BASFOR is based on previous comparisons against observations."

"*p. 8, line 8: Remind us what sigma is again. Is this measurement error?*"

- Sigma is defined in p8 line 3 and is the measurement error

"*p. 8, line 24: "In this study, we made three kinds of calibration". Later, you refer to the multi-site tests with 6 different calibration datasets. Can you clarify here how the three kinds are related to the 6 datasets? Will help reader to follow along later.*"

- The three kinds of calibrations are defined in sections 2.5.1, 2.5.2 and 2.5.3. The calibrations with 6 different datasets are defined in section 2.5.2 as one of the three kinds of calibration.

"*Does the pooled calibration include pine, spruce and deciduous sites? (It seems like it based on page 9, line 14: "identical to the default calibration labelled 'All' above"). Why not restrict this pooled test to just the pine sites? Why did you choose pine for this pooled vs. unpooled set of tests?*"

- We will add text in the revision to make it clearer that the pooled versus unpooled comparison was for pine sites only.
- The choice of pine was pragmatic as this had the greatest number of sites (9) without having to run single calibrations for all 22 sites. The assumption being that the results that we found are robust for the sites that we did not run. We will add this to the revised version of the paper.

"*Page 9, line 27: Please explain what "burn-in" is.*"

- The term "burn-in" is jargon from the MCMC method. We will add text to explain this term in the paper revision.

"*Sections 3.1 and 3.2 (page 10), isn't it pretty obvious that the calibration would bring the posterior closer to the observations than the prior? How representative are the two sites that you chose in terms of the calibration performance of other sites?*"

- It is not obvious that the posterior will be closer to the observations than the prior especially for multi-site calibrations. We show several examples where some of the simulated variables were further away from observation after calibration. This is reported in section 4.1.
- Given the length of the paper we will remove section 3 in the revised version as it is superfluous to the story of the paper.

"*p. 16, line 24: "In general, calibration data was less effective in reducing model-data differences when there were larger inconsistencies present." This is a pretty vague sentence, inconsistencies between what and what?*"

- To avoid the discussion being overly long-winded and cumbersome we use the shorthand term "inconsistencies" which is defined in the text in the previous paragraph. We write: "the single dataset calibrations highlighted where there were inconsistencies either between the different observational datasets or between the observations and the model or both (subsequently referred to in the text as just "inconsistencies")"

"*p. 17, line 18: "Further, large reductions in uncertainty also occurred when the observations included in the calibration had a tendency to increase model-data differences." Shouldn't this be to decrease model-data differences? Shouldn't model uncertainty theoretically go down when model/ data fit improves?*"

- This counter-intuitive result is explained in the text in the next sentence: "While these reduced uncertainties are mathematically correct, if we assume that the model and data do not have structural errors and systematic biases, the small a-posteriori uncertainties are not useful if the models are to be used for predicting the future of European forests, with appropriate uncertainties quantified."

"*p. 19, lines 12-13: "suggesting a strong relationship between the underlying processes as represented in the model." This seems like a self-evident statement that the ecosys- tem variables included in the model are likely correlated with one another. Can you be more specific here?*"

- The key point is that the single-dataset calibrations help to diagnose strong relationships which help to explain the prior to posterior model-data and uncertainty changes found in the multi-dataset calibration. This point will be clarified in the next revision.

Tables and Figures

"*It would be good to reduce the number of figures in the paper, e.g. by combining high and low frequency variables into a single plot (i.e. Figs 5/6, 7/8, 9/10, 11/12), and then indicating high vs. low frequency using background shading or labels. The authors may also consider just showing one metric or another rather than both in the main text (moving the alternative metric to SM). Given a "storyline" in the paper, how does each figure help to illustrate an aspect of this story? The figure captions should also indicate what are "good" values for the metrics being shown, or otherwise, how to interpret them, i.e. that values lower than 1 imply some amount of uncertainty reduction.*

- Figures will be merged in the revised version where this can be done without the information shown being comprised. This is most likely for figures 5-8 but may also be possible for some of the box-plot figures.

"*Figures 1 and 2: how did you choose which sites to show? Shouldn't the posterior be closer to the observations by definition? Maybe combine Figures 1 and 2, and/or move to SM? These figures could also use a legend. Should also specify which calibration is being shown. Is this "all data"?*"

- Section 3 and these two figures will be removed in the revised version of the paper.

"*Figures 3, 4 and 13: It would be helpful to show means for each row and column in these plots, at least across sites for each variable. Figures 3 and 13 are also small and hard to read. For Figure 3, consider flipping each sub-figure and then stacking the two up and down? Might make more space for bigger labels. Again, which calibration are you looking at?*"

- We will increase the size of the plots 3, 4 and 13 (inc. labels).
- We will add that these figures were for the calibration where all the data were included.

"*Figures 9 and 10: where is the spread in the boxplots coming from? Is this the spread across sites? Should specify this in caption, and also that you're looking specifically at pine sites here.*"

- We will add that the spread was due to sites and that the figure refers to pine sites only.

"*Table 3: this is mentioned in text after Table 4, therefore the tables should be re-ordered. Also, please include the full name for each variable, which will help reader to follow along in the text. It would also be nice to show the timescale and the start and end years for each variable. Why is GPP in bold?*"

- Table 3 and 4 numbers will be swapped in the revised version.
- Full names for the output variables will be added.

- The start and end years are different for each site.
- That the GPP data was not included in the calibration is already mentioned in the caption of the table.

"*Table 4: Do the "measurement available" columns refer to the previous columns? Should clarify.*"

- We will clarify which columns the "measurement available" columns refer to in the revised version of the paper.

"*Table 5: What is the sigma value referred to here? Should define. Also, why are Rh, NO2 and NO included here, but not in any of the results?*"

- The sigma is defined in the text but we will add a description for sigma in the caption of the table.

Supplementary material

"*Why is Figure 2 in between Tables 1 and 2?*"

- The caption and table number has been omitted from the first table. This will be corrected in the revised version.

"*Figures 3, 5 and 6: what do you mean by "ancillary observations" in the figure captions?*"

- "ancillary observations" will be changed to "low frequency variables" in the revised version

"*How are these different from ordinary "observations", e.g. in Figure 4 caption?*"

- We will clarify in the caption that the spread of the box-plots are over sites and variables.

Response to Referee 3

"*The authors calibrate the BASFOR forest model using various data and analyze what data best constrains the posterior predictive model uncertainty. The type of work presented is important even though new methodology is not presented. The novelty of the paper comes from using more data and in a more varied setting than was done before. I will comment on the calibration part in these comments, since I'm not an expert of forest modeling. Whereas the work does have merits, it is seriously lacking in detail, and many design choices seem rather arbitrary. There are no formulas describing what was done - even though Bayesian model calibration is a mathematical exercise.*"

- A consistent criticism of the referee is that we do not evaluate and validate the model (BASFOR) and that we do not analyse the influence of the calibration on model parameters. These are in themselves interesting and valid questions and could be the subject of another paper but our paper addressed a different question namely the informativeness of the rich calibration data for model outputs. Many of the referees comments and our responses return to this central point of a mismatch between our stated aims in the paper and the paper that the reviewer would like to see.

The following comments address similar issues so we answer them together.

"*There are several other issues as well, the most important ones being lack of cross validation, model and model parameter descriptions are virtually missing,*"

"*Posterior parameter values are not described, not even in the supplements. That belongs to the main text. I would want to see full descriptionos of the parameters, priors, and posteriors in the main text, and a discussion referring to model equations about what the results of the calibration mean.*"

"*1. One general issue with model calibration is, that once calibration is done, it can be used to inform where the model can be improved - where it does not perform well and where it does not. I don't understand why this aspect is not discussed. Please add this aspect throughout, especially in the discussion.*"

"*5. p 5 section 2.2. You need to give the model equations, pointing out the parameters you are calibrating in those equations. This is crucial - at the moment it is impossible to say what happened in the calibration, when the parameters are not described at all. The manuscript should be self contained in that by reading it one does understand what happened.*"

"*9. p6 l28: Why the covariance structures are not shown? The pairwise posterior marginals contain the most interesting information: What directions in the parameter space are constrained and what are not; are the correlations linear; are the distributions unimodal and close to Gaussian etc. An important part of the analysis should be model parameter (and hence process) identification and finding information about what data constrains what processes. The covariance structure is an important key to this analysis. Please include a figure of the 2d-marginals with probability contours (using e.g. Gaussian KDE), and discuss.*"

"*20. Generally about results: Of course a fitted model fits data better. To know anything of how successful / good the calibration is, a k-fold cross validation (with a suitabke k) should be done instead and all the results should be reported for those. That would give information about how good the calibration is for predictive purposes. At the moment the results read a little like "the model was fitted to data and after fitting, the model fits the data better". With cross validation the results would be significantly more valuable*

*and interesting.*"

"*21. p. 10 l. 18: "Model output uncertainty is reduced from prior to posterior". This is not interesting without the cross validation - the result could be very site-specific and for predictive purposes the result could be in many places different. This being said, there is some value in the analysis, that calibrating the model using data x reduces error in variable y. But without including the model equation this is hard to see. Please, discuss these types of results with references to the model equations. Also, I would like to see what the expected value of the change of this error is.*"

"*23. section 4.1 would be meaningful with the cross validation, but not so much otherwise*"

"*31. sect. 7.1.1 generally: without the equations it is difficult to say much about whether the results are just obvious or if there is something interesting here. The discussion is not very helpful either. Please refer to model equations to explain model behaviour.*"

- The reviewer was interested in assessing BASFOR but the purpose of this paper was not model evaluation and validation for which we agree cross validation would be helpful. Our stated aim was to assess the informativeness of the data in the calibration in term of model outputs. We therefore assessed how model-calibration data differences changed as a result of the calibration.
- As stated in the introduction we focused in this paper on model output variables which are shared across models rather than parameters which are model specific. If the focus of our study had been on evaluation and validation of BASFOR then we would have made an analysis of how model parameterisation changed as a result of calibration.
- The model description is deliberately concise. The focus of this study isn't BASFOR. It is not a new model and has been published before. In the model description we include a reference to a github repository where the full model code is

available and a 33 page user guide, which gives full information on BASFOR for the interested reader.

- The processed based forest-model used in this study is relatively complex (12 state equations, 54 parameters and highly nonlinear). We agree that calibration results can be a useful diagnostic tool to shed light on possible model inadequacy, and discuss this on p17 l23-26. For example, the inconsistencies in the calibrations with NEE and evapotranspiration data implying conflicts in the model representations of the carbon and water cycles. However, for such a complex model it is just not possible to attribute the meaning of results of the calibration to individual model equations in the way that the reviewer suggests.

The following comments address the same issue and will be answered together.

"*and although Bayes' rule is described, a probabilistic explicit observation model is not given. Details of the MCMC experiment are not properly described either.*"

"*13. p7 l31 "uncertainty about random data error", do you mean "data uncertainty". The data uncertainty should be discussed more widely, see later.*"

"*14. p. 8 l. 3. What is the observation equation (like yt = M(x0,t;theta) + eps, where eps ~ some distribution)? How does the averaging come into play in that equation? And again: "measure of random error about ith data point". Do you mean measurement error? Or something else? Please discuss the merits of the chosen likelihood function, what does it mean? (also see next item)*"

- "uncertainty about random data error" is uncertainty about error in measurement.
- The probabilistic explicit observation model is given in equation 2 in the paper.
- We already discuss the merits of the chosen likelihood function given in Eq 2. on p7/8 as "The uncertainty about random data error has often been represented

by independent Gaussian distributions for the observations. As discussed in Van Oijen et al. (2011) this can overestimate the information content of the observations leading to an underestimate in uncertainty. To help alleviate this issue we used the heavy-tailed distribution of Sivia (Sivia and Skilling, 2006)."

"*The manuscript is too long - there is a lot of repetition, and many of the conclusions are quite obvious. Such repetition should be reduced to a minimum. In addition to this, there are still very obvious editing errors, which need to be corrected. Not all of them are listed below, since there are too many.*"

- We agree that there is material included in the paper that is not essential and detracts from the main story for the paper. As detailed in our response to reviewer 2 we will remove non essential material from paper in the revised version.

"*2. Please make the abstract more concise and articulated: what's new in the work and why it is important. At the moment it is a little long and unclear. And please check the language/style.*"

- We agree that the abstract can be usefully shortened while retaining the main points that we wish to make.

The next two comments address a similar issue. Therefore, we will answer them together.

"*3. One enormous source of uncertainty is biases in input data. Could you please comment on this aspect.*"

"*7. p 5 l 22: planting density assumption: yet again an important source of potential bias / uncertainty that is not discussed at all. Please include in a discussion of the input data uncertainty.*"

- The referee was correct to suggest that the novelty of this paper was the richness of the calibration data. However, we only had access to one set of model inputs in this study. Therefore, our focus was on the impact of this rich dataset on the calibration rather than a quantification of all the possible uncertainties (including uncertainty in the inputs).

"*4. p 4 l 30: The 1m rooting depth seems pretty arbitrary, and I'd guess it does affect the results. Why was this value chosen, and if it's not known, why isn't it included in the calibration? How much are the results affected?*"

- The heading of this section (2.1.3) was "Soil depth and soil water retention data used to set prior ranges." and later in section 2.4.2 we make clear that rooting depth is calibrated. However, we accept that section 2.1.3 could be improved to make it clearer that the data described here are used to set prior modes and ranges.

"*6. p 5 l 16: "replicated backwards. . . ?" what does this mean? Please describe in a detailed and compact way what you did, preferably with equations.*"

- We have answered this question in our reply to reviewer 1.

"*8. p6 l24: What parameters where used for the Beta distribution? Why? How did you use literature data to obtain the priors? Tables in supplements should be moved / summarized in the main text, with references to how the priors were chosen.*"

- The parameters that we chose were for an uninformative Beta distribution (Van Oijen et al 2013). This reflects our uncertainty in the parameters prior to the calibration. We will add these details to the revised version of the paper.

- As stated in the text the parameter ranges were chosen from a combination of measured data sections 2.1.3, 2.4.2 and 2.4.3 and experience from previous published studies with BASFOR (section 2.4.2) "Where possible, parameter priors were estimated from measurements at sites or from literature. In many cases this was not possible as either data were not available or the model parameter had a different role in the model than that measured. In these cases a wide prior was used. Values for these wide priors were guided by previous studies with BASFOR (Van Oijen et al., 2005; Van Oijen and Thomson, 2010; Van Oijen et al., 2011)."
- Further details of how parameters were estimated from "measurements at sites or from literature" will be added in the revised version.

"*10. p7 l16: In a manuscript describing Bayesian calibration of a model, the prior values most definitely belong to the main text, not to the supplement. Please include it here. I'd like to read the prior values for different forest types from a table. Justification of the prior distributions used should be included.*"

- As stated above our focus in this study was not on the Bayesian calibration of a model and we deliberately chose not to clutter the paper with information about parameters when the focus of the study is the informativeness of the data included in the calibration for the model outputs.
- We explained our choice of prior in the text: "We chose the Beta distribution for the prior probability distribution, which is bounded and can be non-symmetric.

"*11. p7 l26 "Prior uncertainty was set at 20% of mode value..." But isn't your "prior uncertainty" set by the parameters of the Beta distribution? Also, in any case the 20, 30 and 40% values are arbitrary, and this should be stated explicitly as a potential source of error. Please clarify.*"

- Yes these ranges are used in setting the alpha value of the Beta distribution. The beta value is 6. - alpha. For clarification we will add these details to the paper.
- 20, 30 and 40% values are not arbitrary but are based on the expert judgement of those who collected the measurements. We will add text to this effect in the revised version.

"*12. p7 l30 ..."at a given parametrisation" should rather be "at a given point in the parameter space" or something along those lines*"

- We will change the text as suggested.

The next two comments address a similar issue. Therefore, we will answer them together.

"*15. (same) It looks to me like you are treating the residuals as independent. Is this a reasonable assumption? An alternative would be to fit a time series model (like an AR/ARMA/ARIMA etc model or such) to produce wider posteriors. You should analyze the residuals and verify that whatever probabilistic observation model description you are using, your residuals in the end conform with your error model (please show histograms of the residuals and autocorrelation functions in the supplements). If you are not able to do this, you must aknowledge that all the ranges in the figures and the scaling ("steepness") in the posterior probability distributions is arbitrary. At this point, what you have left, is then the covariances of both parameters and predicted errors / quantities, and discussing those would still be valuable. In the best case scenario, you should add both.*"

"*35. p. 19 l. 27. It is unclear to me how model structural errors were represented in your work. How did you quantify that "uncertainty is not reduced inappropriately*"

- In common with many Bayesian calibrations of complex process based models we are treating the residuals as independent (see the discussion below).
- The reviewer raises an important point about the influence of model and data systematic errors which is not the focus of the paper but will influence the results as discussed in section 7. The results suggest inconsistencies between the model and the calibration data but it is not straightforward to attribute these to model errors, data errors or most likely both. Common with many calibrations of processed based ecological models we do not include terms in the calibration that represent model structural errors and data systematic biases. This omission would be especially important if our goal was a definitive quantification of all the uncertainties present. In ignoring these sources of uncertainty it is likely that our quantifications of uncertainty are smaller than they would be if they were included. Unfortunately representing model structural error for complex processed based models such as BASFOR is not straightforward especially since, as we discuss in the paper, the data are also likely to have systematic biases. Nevertheless, we agree (as stated in the conclusion) that the inclusion of terms that represent model structural and data systematic error should be the next important step for the calibration of process based forest models.

"*16. p8 l7: So you are predicting monthly values? Clarify and see previous point.*"

- Model outputs are aggregated to 30 day averages.

"*17. section 2.4.5. Details are missing. What was the proposal distribution for MCMC? Where is the picture of the chain? (please include). How long was the burn-in? What was the acceptance ratio? Effective sample size? And was there a reason to go with just Metropolis? Usually e.g. Adaptive Metropolis works a lot better (unless if you happen to a priori know what the optimal proposal is)*"

- The proposal distribution was multi-variate normal and the step length was proportional to the standard deviation of the Beta priors. We will add these details to revised version of the paper.
- This paper involved 54 separate MCMC calibrations. This is too many to provide pictures of the chains for the 54 parameters (54*54) and information on burn-in, acceptance ratio, effective sample size etc and would swamp what is already a long paper with information that would not have wide interest.
- As stated in the section 2.4.6 each separate MCMC was checked for convergence using a Gelman-Rubin statistic.
- Adaptive Metropolis is not proven to be better than Metropolis for high dimensional process-based models (see forthcoming publication by Minunno et al).

"*p8 l. 24. The lonely sentence could be left out. Throughout the text there are lots of these types of sentences, and many of them could be just removed.*"

- We will remove extraneous sentences that do not add to the clarity os the paper in the revised version.

"*19. p 10 l1. I'd really like to see the corresponding contours on the pairwise 2d marginals instead / in addition.*"

- We agree that correlations between the output variables are important. We will include a new plot showing these for the calibration that includes all the data.

"*22. p. 11 l. 2 Are you sure this is the right metric? What if you get some really bad outliers but mostly good behavior? Is this then an acceptable model? If such behavior never occurs, the approach can be ok.*"

- Quantiles are known to be robust to the influence of outliers.

"*24. p. 12 l. 1-2: I have difficulty understanding the sentence.*"

- This sentence will be reworded in the revised version of the paper.

"*25. Section 5.1. Would love to see what these results mean with some short and concise discussion. A time series of the predicted values before and after calibration would be valuable.*"

- Two examples of time series are indeed shown in section 3 of the paper. With 22 sites and 11 variables and six experiments (22*11*6) for both prior and posterior there are too many time-series plots to include all of them in the paper. That is why we chose to show summaries in for example fig 3, 4 and 13.

"*26. General: the observation set is advertised to be "rich" a few times too many*"

- We respond to this point in our response to referee 2.

"*27. p 15 l. 20. In practice there are always inconsistencies with real data. I'd say such results tell about weighting of the different variables in the likelihood function (choice of error model).*"

- The key point here is that the weights of the data in the likelihood function were not chosen arbitrarily but p8 l12 "are informed by knowledge of how the data were collected and also by literature".

"*28. p. 15 l. 21, 25: We will return. . . & We will discuss. . . could compactify the text a little here and leave these out. p. 16. Lots of text for the content. Simplify and remove repetition.*"

- The text will be shortened in the revised version by for example removing phrases such as this comment suggests.

"*29. p. 15 l. 28 "mathematical probabilities", rather say "a probabilistic model" and describe the model properly as I mentioned earlier.*"

- This wording will be changed in the revised version.

"*32. p. 18 l. 32. You recommend using a hierarchical model for describing the parameters, but you are not doing a hierarchical modeling yourself. Why? Also, if you were to hierarchically model the parameters, you'd need to have a way of predicting the parameters for predictive purposes. How would you do that?*"

- As the referee suggests, hierarchical modelling is not straightforward for a model of this complexity. It is an area of active research for us but we are not yet able to make this analysis.

"*33. p. 19-20. I would like to see a more compact Conclusions section, and also I'm not sure that the bullet point style is a good idea.*"

- Bullet points give a clearer summary of the main conclusions of the paper than a paragraph of text so we disagree with this comment.

"*34. p. 19 l. 22. What's the role of the prior?*"

- We chose wide priors for this study (see above) reflecting our uncertainty in the parameter priors. Therefore the priors do not overly constrain the calibration.

"*Fig. 2: The wavy shapes look strange – where did that come from?*"

- These "wavy shapes" reflect seasonality (trees are more active in summer than in winter).

"*Figures: Merge 5&6, 7&8, 9&10, 11&12, 14&15; check legend style/formatting and maybe optimize the general presentation, with improved captions. Explain in words what is seen in pictures.*"

- Figures will be merged in the revised version where this can be done without the information shown being comprised. This is most likely for figures 5-8 but may also be possible for some of the box-plot figures.

Technical corrections

"*grammar: e.g. p1/l12 inclusion. . .were => was p4 l 29 www address to references? (check journal style, but it does not look good.)*"

"*p6 l6 on: sentences don't have verbs.*"

"*p6 l19: "probability calculus" => Bayes' rule*"

"*p8 l 26 "of" missing most likely p13 l30 2nd sentence, sounds a little strange.*"

- These corrections will be made in the revised version of the paper.